# Innovative Pre-Clinical Data Using Peptides to Intervene in the Evolution of Pulmonary Fibrosis

**DOI:** 10.3390/ijms241311049

**Published:** 2023-07-04

**Authors:** Karina Smidt Simon, Luísa Coutinho Coelho, Paulo Henrique de Holanda Veloso, Cesar Augusto Melo-Silva, José Athayde Vasconcelos Morais, João Paulo Figueiró Longo, Florencio Figueiredo, Leonora Viana, Ildinete Silva Pereira, Veronica Moreira Amado, Marcia Renata Mortari, Anamelia Lorenzetti Bocca

**Affiliations:** 1Department of Cellular Biology, Institute of Biological Sciences, University of Brasilia, Brasilia 70910-900, Brazil; karina.smidt.simon@gmail.com (K.S.S.);; 2Laboratory of Respiratory Physiology, Medical School, University of Brasilia, Brasilia 70910-900, Brazil; camelo@me.com (C.A.M.-S.);; 3Hospital of the University of Brasilia, University of Brasilia, Brasilia 70910-900, Brazil; 4Department of Genetics and Morphology, Institute of Biological Sciences, University of Brasilia, Brasilia 70910-900, Brazil; 5Laboratory of Pathology, Medical School, University of Brasilia, Brasilia 70910-900, Brazil; 6Department de Physiological Sciences, Institute of Biological Sciences, University of Brasilia, Brasilia 70910-900, Brazil

**Keywords:** idiopathic pulmonary fibrosis, inflammatory response, immune modulator peptides, peptide ToAP3, peptide ToAP4

## Abstract

Idiopathic pulmonary fibrosis (IPF) is a progressive, relentless, and deadly disease. Little is known about its pathogenetic mechanisms; therefore, developing efficient pharmacological therapies is challenging. This work aimed to apply a therapeutic alternative using immunomodulatory peptides in a chronic pulmonary fibrosis murine model. BALB/c mice were intratracheally instilled with bleomycin (BLM) and followed for 30 days. The mice were treated with the immune modulatory peptides ToAP3 and ToAP4 every three days, starting on the 5th day post-BLM instillation. ELISA, qPCR, morphology, and respiratory function analyses were performed. The treatment with both peptides delayed the inflammatory process observed in the non-treated group, which showed a fibrotic process with alterations in the production of collagen I, III, and IV that were associated with significant alterations in their ventilatory mechanics. The ToAP3 and ToAP4 treatments, by lung gene modulation patterns, indicated that distinct mechanisms determine the action of peptides. Both peptides controlled the experimental IPF, maintaining the tissue characteristics and standard function properties and regulating fibrotic-associated cytokine production. Data obtained in this work show that the immune response regulation by ToAP3 and ToAP4 can control the alterations that cause the fibrotic process after BLM instillation, making both peptides potential therapeutic alternatives and/or adjuvants for IPF.

## 1. Introduction

Pulmonary Fibrosis (PF) results from tissue damage or interstitial pulmonary disease [1]. When the etiological agent is unknown, it is called idiopathic pulmonary fibrosis (IPF). IPF is a severe form of PF found in adults, limited to the lungs; it has little effective treatment, involves the complexity of the health system, and has high morbidity and mortality rates [2,3,4,5,6]. Little is known about the mechanism that leads to the fibrotic process in IPF because there is no prominent inflammatory process at the initial stages of the disease but an excessive apoptosis rate of epithelial alveolar epithelial cells (AECs) [7,8]. However, inflammatory responses are associated with the initial stages of IPF [9,10]. Pulmonary injury induces immune cell migration, supporting an initial inflammatory response with high TNF-α, IL-10, and IL-1β release. These mediators can induce AEC apoptosis, fibroblast growth, increased collagen production, and the production of growth factors such as TGF-β [11,12,13]. TGF-β is capable of inducing AEC apoptosis and, together with TNF-α, induces the transformation of fibroblasts into myofibroblasts, cells that, combined with epithelial damage, are a feature of IPF [7,14].

Many therapeutic alternatives and protocols have been developed over the years to treat IPF. However, little has been achieved towards a cure or improving quality of life. Although many studies have shown the role of cytokines in the inflammatory response and their importance in experimental models of IPF, most therapeutic protocols that block exacerbated lung inflammation failed in pre-clinical stages [15,16,17]. These results have led to the belief that IPF development is not closely related to the immune response (IR). Because of the dissociation between the fibrotic process in IPF and the IR, new research into treatment for this disease focuses on antifibrotic approaches such as nintedanib and pirfenidone [16,17]. Nintedanib acts by inhibiting the tyrosine kinase pathway, decreasing the proliferation, migration, and survival of lung fibroblasts and myofibroblasts, and reducing extracellular matrix deposition [18,19]. Pirfenidone’s activity is poorly understood, but it seems that one of its mechanisms of action is the inhibition of myocardin-related transcription factor (MRTF) activation, decreasing the cells’ mechanical stress, which might contribute to its therapeutic effect [19,20,21]. The use of Nintedanib and Pirfenidone has significant limitations, only slowing disease progress and showing little efficacy for patients in more advanced stages [16,17,21]. Furthermore, they are high-cost drugs and cause side effects that often lead patients to discontinue use [17]. Thus, studies are still needed to develop new pharmacological alternatives for the more efficient control of IPF. Currently, the only available cure for this disease is lung transplantation, which, in addition to the surgical process risk and the low survival of patients in the first two years after transplantation, is not recommended for most cases of IPF due to the age range of most of the patients and the comorbidities associated with the disease [22].

Although not the main trigger of the fibrotic process in IPF, there is an understanding that the immune system is responsible for maintaining the continuous fibrotic process [13]. Thus, considering that tissue damage occurs due to the dysregulation of the mechanisms responsible for tissue healing during the local inflammatory process, using regulatory elements of the immune response at the beginning of the fibrotic process can be an attractive therapeutic alternative. Therefore, pharmacologic therapies based on the mechanism of antimicrobial peptides (AMPs), such as the *Tityus obscurus* peptides ToAP3 and ToAP4, which have anti-inflammatory potential and regulatory properties in cytokine production [23] could have therapeutic potential. These peptides were characterized as capable of regulating the secretion of pro-inflammatory cytokines through interaction with Toll-like receptor 4 (TLR4), interfering with the expression of class II MHC in DCs. They also have low cytotoxic and hemolytic activity, characteristics that make them good candidates for therapeutic use for diseases whose maintenance and progression are related to the IR, such as IPF [23,24]. In this work, we aimed to investigate the potential of ToAP3 and ToAP4 as potential pharmacological alternatives or adjuvants for IPF treatment by characterizing the inflammatory response of bleomycin (BLM)-induced BALB/c mice in an IPF murine model.

## 2. Results

### 2.1. The Animal Model Strategy Showed Long-Term Inflammation with Lung Fibrosis Formation

The major IPF experimental model is bleomycin-treated C57bl6 mice [12,25,26]. However, these animals develop acute fibrosis, which leads to self-healing 21 days post-instillation (p.i.) of BLM, making it difficult to analyze the efficacy of the new drug. We used BALB/c mice to carry out long-term tissue inflammation that could mimic PF. They used to be classified as resistant to PF [27,28], but in several experimental lung infectious diseases, they develop chronic inflammation with fibrosis formation, as in the murine model of paracoccidioidomycosis [29,30] and histoplasmosis [31,32].

Before using peptides, we ratified the murine model, following the animal survival p.i., and verified that the number of lost animals was minimal, showing that BALB/c mice are more likely to survive after BLM instillation (Appendix A). Furthermore, we did not see a significant weight loss in these animals during the experiment (Appendix A).

We also analyzed tissue morphology alterations and pulmonary ventilatory mechanics. After BLM instillation, we followed the fibrotic process development in BALB/c mice for 60 days, noticing the most prominent changes were detected at 5, 30, and 60 days p.i. compared with non-instilled mice. The pulmonary parenchyma of these animals at 30 days p.i. appeared slightly enlarged, and it was possible to identify dilated capillaries with their lumen filled with red blood cells. The presence of different-caliber airways was also verified, represented by structures with walls constituted by fibromuscular tissue and internally lined by respiratory pattern ciliated epithelium. At 60 days p.i., the cellular infiltration reduced, and fibrosis formation prevailed, with a considerable area of collagen deposition (Appendix A).

The lungs of live mice were analyzed by µCT images, showing indicatory areas of fibrosis (Appendix A) and collagen deposition (Appendix A). µCT analysis showed increasing opacity of the lung parenchyma of BALB/c mice after BLM instillation since the 5 d.p.i. until the endpoint of the analysis. These pronounced diffuse opacifications progressed to zones with higher density in the lungs, suggesting tissue damage. To confirm the fibrosis formation by specific staining, we used immunohistochemistry techniques (IHQ) to identify collagen types I, III, and IV and their respective quantifications to compare lesion regions. We saw increased deposition of collagen types I and III until 60 d.p.i., and collagen type IV had significant accumulation in some regions until 60 days p.i. (Appendix A). Collagen I and III deposits result from an unregulated wound-healing process initiated after tissue injury, especially in the late stages. The collagen deposition confirms the µCT images, with an increase in collagen type I and Type III deposition and a reduction in Type IV deposition, confirming the worse progression of the model (Appendix A).

The IPF was evaluated through disturbances in respiratory mechanics. Analyzing the respiratory system resistance (Rrs), there was a significant difference between non-instilled and BLM-instilled mice after 30 days p.i. (about 95%) (Appendix A). Likewise, the elastance (Ers) was significantly modified after BLM intratracheal installation. After 5 days of BLM instillation, Ers increased by about 54% compared with non-instilled mice, with a slight decrease but still 40% higher than non-instilled animals. On the 30th day p.i., there was a peak in Ers, with an increase of nearly 72%. Similarly to Rrs, we observed a significant increase in Newtonian resistance (Rn) only after 30 days of BLM instillation (about 170%). We also observed significant differences between instilled and non-instilled animals in tissue damping (G), with an increase of 57% in the measurements of this parameter on the 5th day p.i., 89% on the 15th day p.i., and 105% on the 30th day p.i.; tissue elastance (H) increased in comparison with control animals at the rates of 90% on the 5th day p.i., 38% on the 15th day p.i., and 78% on the 30th day. After 60 days of BLM instillation, the only parameter that was significantly different in the untreated mice was G. These measures were nearly 40% higher in BLM-instilled mice than in control mice.

These data indicate that there was a significant BLM-induced mechanical impairment of the respiratory system and that this is a pathological characteristic associated with structural tissue changes, so we used this model to mimic the initial stages of fibrosis formation.

### 2.2. The Treatment of Important Morphological Parameters That Contribute to Fibrosis Formation in Lung Tissue

After the lung injury, many stages in the tissue repair process occur and contribute to non-functional healing. We started evaluating some morphological parameters after the treatment with the peptides. According to the histological analyses, the initial inflammation is seen at 5 d.p.i.; therefore, we started the intranasal treatment with ToAP3 and ToAP4 peptides at this point. They were administered at an interval of 3 days, and on the 30th day p.i., we analyzed the pulmonary parameters. There were alterations in the morphological structure between the treated and untreated groups on the 30th d.p.i., with a lower cellular infiltration, more preserved tissue structures, and lower collagen deposition (Figure 1). Considering that collagen deposition has shown differences in the peptide-treated group, we aimed to identify the differences in collagen type in each group (Figure 2A). There was an increase in collagen types I and III after 30 d.p.i. but a decrease in treated groups (Figure 2B). The collagen type IV was increased after the ToAP3 treatment, showing an improvement in basal structural maintenance (Figure 2B).

Likewise, we followed the development of fibrosis in live animals, analyzing the microcomputed tomography (µCT) images of BLM-instilled BALB/c mice with or without ToAP3 and ToAP4 treatment. This brings us closer to the analysis that patients are submitted to, which gives the diagnosis of IPF, allowing translational science. To better understand the μCT data, the frequency of the voxels of interest (VOIs) on the Hounsfield scale (HU) confirmed this tissue damage during the time analyzed. It was noted that 140–500 HU frequency was much higher at 30 days p.i. in the non-treated group (Figure 3B,C) when compared with non-instillation mice (Figure 3A). These results suggest that, as the collagen deposits occurred after BLM instillation, the alveolar space diminished and the density of the tissue increased (Appendix A), leading to respiratory disability. The frequency values of VOIs between −200 and 140 HU found in µCT images were higher after treatment with both peptides in BLM-instilled mice when compared with untreated mice at the 5th and 30th d.p.i., showing a better integrity of the tissue. Moreover, the frequency of VOIs between 140 and 500 HU found in µCT images was lower in treated groups at the 30th day p.i. (Figure 3E,F). The less dense HU quantification (−200 to 140 HU) represented tissue closer to a healthy cofibroticle, and the higher values (140 to 500 HU) represent lung tissues with some kind of solid deposition that was later identified as the fibrotic tissues in the histological analysis. Comparing the percentual of VOIs among the groups, the treated group showed higher levels of −200 to 140 HU and lower levels of 140 to 500 HU (Figure 3I).

Once we had noticed the peptide’s capacity to prevent the course of fibrosis, we wanted to know if, on stopping the treatment, the fibrotic process would continue or if it would be permanently eliminated. After following the same treatment protocol, mice were kept in observation for 30 more days and scanned for evaluation of pulmonary tissue. Their frequency of VOIs between 140 HU and 500 HU and the percentage of VOIs between −200 HU and 500 HU showed no statistical differences compared with animals at the beginning of the treatment (Figure 3B), demonstrating the interruption of tissue damage. These groups of animals had the same time post-instillation as the group at 60 d.p.i. When we compared the three groups, we observed similar results with increased levels of ToAP3 treatment between −200 and 140 and lower levels between 140 and 500 HU. The ToAP4 treatment showed similar levels of HU between −200 and 140 than 60 d.p.i. but higher levels between 140 and 500 HU (Figure 3I).

### 2.3. Peptide Treatments Prevent the Worsening of Pulmonary Mechanical Parameters

The ventilatory mechanics’ profiles were verified in peptide-treated mice (Figure 4). For peptide-treated mice, 5 days p.i., we started the treatment of two groups of mice, in which one was given ToAP3 and the other ToAP4. Rrs showed a reduction of 30% for the ToAP3 group and about 35% for the ToAP4 group when compared with BLM-instilled mice after 30 days p.i. (Figure 3A). However, both groups still had higher Rrs than on the 5th day p.i. As in the morphological studies, we also evaluated the respiratory mechanics of mice 30 days after the end of the peptide treatment. ToAP3 peptide-treated mice showed Rrs 35% lower than untreated animals after 30 p.i. Furthermore, we also observed that in ToAP4-treated mice, Rrs decreased by about 23% 30 days after stopping the treatment (Figure 4A). For the Ers parameter, we observed an improvement of nearly 70% for ToAP3-treated mice and almost 50% for ToAP4-treated mice 30 days after the BLM instillation (Figure 4B). However, 30 days after treatment interruption, there was a natural decrease in the Ers levels (Figure 4B). When we looked at the Rn parameter, we observed an increase in measures for the Rrs parameters in our model; we also saw a similar rise in Rn. Moreover, a significant difference between peptide-treated and untreated animals was noted. However, in peptide-treated animals, we found a decrease of nearly 45% for ToAP3-treated mice and almost 60% for ToAP4-treated mice (Figure 4C). Although measures were still higher than at the beginning of treatments with ToAP3 and ToAP4, there was a significant decrease in Rn 30 days after we stopped the peptide treatment. These differences were even more marked, at approximately 23%, for the ToAP3 peptide (Figure 4C). As Rn represents the resistance of the central or conducting airways, its decrease means that the treatment with both peptides left a lasting improvement in lung function for the inflammation that would lead to fibrosis. For the G parameter, related to the dissipation energy of the alveoli and closely linked with resistance, whose increase reflects a worsening in the respiratory capacity, we observed a gradual rise in G until the 30th day for the untreated group. However, after the treatment with both peptides, we observed a significant G decrease between mice in the untreated and the peptide-treated groups, with approximately 20% for ToAP3 and about 30% for ToAP4. Mice treated with ToAP4 also showed a 10% lower G than the group of mice 5 days after BLM installation (Figure 4D). After 30 days of the last round of treatment, peptide-treated animals maintained a significantly lower G than untreated animals on the 30th day, a significant difference from the last day of treatment.

Still analyzing alterations in the lung parenchyma, H is a parameter that reflects the conservation energy of the alveoli. Therefore, the higher the value of H, the greater the energy required for tissue expansion in the alveoli, and this phenomenon may be related to inflammation and fibrosis. In the evaluation of H, it was possible to observe values around 35% lower than those found for untreated animals only instilled with BLM (Figure 4E). The same pattern was observed in animals 30 days after stopping the treatment.

The treatment did not recover the lung mechanics parameters as in non-instillation mice but reduced these parameters when compared with 30 d.p.i., and, considering the data shown until now, the treatment reduced the tissue cellular infiltration and collagen depositing that can be responsible for the μCT opacity.

### 2.4. ToAp3- and ToAP4-Treatments Modulate Fibrosis Formation through Different Pathways

We followed the histopathology analyses and chose 15 and 30 days p.i. to compare the gene transcript at the beginning and the late point of fibrosis formation. Compared with non-instilled mice, at 15 days p.i. (Table 1), there was an increase in the levels of integrin beta 6 (Itgb6) mRNA and a decrease in the mRNA levels of the homolog of MAD 4 (Drosophila) (Smad4) and the signal transducer and activator of transcription 6 (Stat6). After 30 days of BLM instillation (Figure 5A and Table 1), mice presented an upregulation of the profibrotic genes Fas ligand (TNF superfamily, member 6) (Fasl) and Itgb6 and a downregulation of endoglin (Eng), which is part of the TGF-β superfamily. In addition, there was an upregulation of the matrix metalloperoxidase 13 gene (Mmp13), which is collagenase 3 expressed during chronic inflammation.

For animals treated with ToAP3 and ToAP4, the transcript accumulation was compared with that found in non-treated animals on the 30th day p.i. (Figure 5B and Table 2). The ToAP3-treated group showed several changes in transcript levels. The treatment with ToAP3 upregulated the profibrotic genes chemokine (motif c-c) ligand 11 (Ccl11), the cytokine IL-1 alpha (Il1a), and the cytokine IL-1 beta (Il1b) genes. The most significant genes regulated with this treatment were downregulated, like the profibrotic genes proto-oncogene Thymoma Viral genes 1 (Akt1) associated with the fibrogenesis; alpha V integrin (Itgav), integrin beta 6 (Itgb6); matrix metalloperoxidase 14 (Mmp14) and matrix metalloperoxidase 2 (Mmp2); platelet-derived growth factor, alpha (Pdgfa), serine peptidase (or cysteine) inhibitor, clade E, member 1 (Serpine1), serine peptidase (or cysteine) inhibitor, clade H, member 1 (Serpinh1) all associated with the fibroses formation; transforming growth factor, beta 1 (Tgfb1), transforming growth factor, beta 2 (Tgfb2), transforming growth factor, beta 3 (Tgfb3), transforming growth factor receptor, beta 1 (Tgfbr1), transforming growth factor receptor, beta 2 (Tgfbr2). Furthermore, for all these downregulated profibrotic genes, treatment with ToAP3 downregulated the homolog of MAD 6 (Drosophila) (Smad6) and the tissue inhibitor of metalloproteinases 2 (Timp2), associated with signal transduction and extracellular matrix, respectively. The treatment with ToAP4 downregulated only the profibrotic genes chemokine receptor (motif c-c) 2 (Ccr2), Serpine1, and Smad4.

The PCR array included several cytokines associated with fibrosis formation, such as IL-1α, IL-1β, IL-4, IL-5, IL-13, IL-10, TNF-α, and TGF-^®^. Although none were differentially modulated in BLM-instilled mice, we analyzed some cytokines to confirm their participation in the fibrosis formation process. In BLM-instilled mice, we observed increased production of TNF-α, IL-1β, IL-13, IL-10, and TGF-β until the 30th day p.i. in all animals analyzed (Figure 6). We found that IL-10 had reached its highest levels in the first 5 days after BLM instillation and kept high levels for the first 15 days p.i. (Figure 6C). After that, IL-10 levels gradually declined. The treatment with ToAP3 did not modulate TNF-α release (Figure 4A) but decreased all the other cytokines. For the next 30 days, we observed that the results varied with increasing levels of IL-1β (Figure 4B) and reduced levels of TGF-β (Figure 6E). The treatment with ToAP4 increased TNF-α and TGF-β (Figure 6A,E) and decreased IL10, IL-13, and IL-1β (Figure 6B–D). For the next 30 days, after treatment interruption, this group showed similar levels of cytokines measured, except for TNF-α levels, which were significantly lower than in the untreated animals (Figure 6A). In contrast, IL-1β levels increased after stopping the treatment but did not reach the same levels at the beginning of intranasal administration (Figure 6B). So, the treatment with peptides decreases the levels of cytokines associated with fibrosis formation and keeps them at a lower level.

## 3. Discussion

PF is a progressive and debilitating disease that is commonly refractory to available treatments and presents a broad spectrum of clinical manifestations [33]. Its formation is associated with predisposing factors and several agents that establish this disease in contact with the lung [34]. Its highly aggressive form, IPF, leads to high-cost hospitalizations and a high mortality rate [6]. The use of two antifibrotic drugs, nintedanib and perfenidone, was a game-changer for IPF therapeutics. However, they are expensive drugs that can only delay the disease’s course and provoke strong collateral effects, which are usually responsible for the discontinuation of treatment [4,35]. Therefore, developing new pharmacological alternatives for IPF treatment is still necessary. Although the immune system may not be the main trigger of the fibrotic process in IPF, there is an understanding that it is responsible for maintaining the fibrotic process [13].

To better understand the evolution of IPF, many animal models have been used throughout the years, including bleomycin (BLM)-induced PF. The literature proposes that BALB/c mice resist the BLM-induced PF; however, this model has been used lately with good results [27,28,36]. Therefore, here we make a brief description of this IPF murine model throughout the disease’s development, carrying out respiratory function associated with immunological effects and tissue remodeling, and evaluating the impact of intranasal treatment with two peptides, ToAP3 and ToAP4, which were described as capable of regulating inflammatory cytokine production [23].

IPF is characterized by a reticular pattern with basal and peripheral predominance, peripheral traction bronchiectasis and bronchiolectasis, and a honeycombing pattern in high-resolution computed tomography (HRCT) caused by collagen deposits on lung tissue [3]. Indeed, in our experiments, µCT analysis showed an increasing opacity in the lungs of BALB/c mice during the 60 days following BLM instillation. Collagen I and III deposits result from an unregulated wound-healing process initiated after tissue injury. Deposits and morphological alterations have been described in many species other than humans over the years [27,37,38]. They also occur due to the damage caused by the inflammatory process and its mediators, which are part of the induction and progression of IPF [39]. After 60 days of BLM instillation, collagen deposits seen in histopathological analysis corroborate the opacity in µCT images and the percentage of VOIs found in these exams, but they also suggest that although fibrosis persists, the inflammatory process decreases. We also observed that the regions of opacity seen in µCT images of ToAP3 or ToAP4-treated mice were much smaller in contrast to those observed in mice 5 days p.i. and, in untreated mice 30 days p.i., whose lungs presented large areas of opacification and a higher frequency and percentage of VOIs from 140 to 500 HU. These results suggest less collagen accumulation in the lungs of mice treated with ToAP3 or ToAP4 and indicate that both peptides regulate inflammation in a manner consistent with their characteristics described in the literature and, consequently, in the development of fibrosis [23].

The respiratory mechanical disturbances observed in BLM-instilled mice are consistent with what was previously reported by other studies that demonstrated interstitial lung fibrosis and lung remodeling with increased collagen deposition [40]. Increased Ers and increases in the viscoelastic mechanical parameters G and H following BLM instillation arise from distortion caused by the inflammation of lung tissue, structural modifications of the extracellular matrix, and surfactant dysfunction associated with progressive alveolar de-recruitment [39,40,41]. It is interesting to note the increase in Rn 30 days after the intratracheal instillation of BLM. This result is in line with Polosukhin et al.’s findings, who demonstrated increased Rn and airway remodeling in mice with prominent peribronchial fibrosis 4 weeks after tracheal instillation of BLM [42]. They also observed partial resolution of airway remodeling after 8 weeks of administration of BLM, which could result from the alterations in the inflammatory process. We observed in the morphological analysis that on the 30th d.p.i., there is an inflammatory process in the lungs of BLM-instilled mice with gradual collagen deposition. It was assumed that the regulation of this inflammation by the peptides used in this work would decrease the cellular infiltrate, which hinders the ventilatory mechanics and controls the fibrotic process and the establishment of a chronic inflammatory process.

Morphological analysis and ventilatory mechanics confirmed our hypothesis that immunomodulatory elements would regulate the inflammatory and scarring processes. Therefore, we aimed to explore the possible alterations in immune mediators related to IPF. Cytokines are immune mediators with inflammatory and anti-inflammatory functions that can display fibrotic and antifibrotic roles in disease models. One crucial pro-inflammatory cytokine that appears at the beginning of the unregulated inflammatory process that leads to IPF is TNF-α; in the lungs, it is mainly a macrophage product after BLM exposure [43,44]. The presence of this cytokine not only contributes to tissue damage but also, even in nontoxic concentrations, stimulates fibroblast growth and promotes the differentiation of pulmonary mesenchymal cells into myofibroblasts. In addition, the increasing levels of TNF-α mRNA in the lungs after stimulation with BLM are associated with the fibrotic process [11,43,45,46,47,48,49]. This process occurs through the activation of the NF-κB signaling pathway, which, together with TGF-β, is capable of inducing type II alveolar cells to change their morphology to a fibroblast-like cell that expresses high levels of α-SMA associated with fibrils [11,14,43,46,49]. Although the idea of blocking TNF-α seemed promising, this approach was seen to have little effect on IFP patients and was detrimental to individuals with rheumatoid arthritis who have PF [38,43,46,50,51,52]. Here, we observed increased production of TNF-α with high levels of this cytokine on the 30th day p.i., and these data are compatible with what we observed using the PCR technique when the highest levels of NFκB transcript after instillation were presented on the 30th day p.i. When treating the mice with ToAP3 and ToAP4, few changes in TNF-α levels were observed compared to those in the BLM-instilled mice. TNF-α levels were similar for ToAP3-treated mice even 30 days after treatment interruption. However, ToAP4-treated mice presented significantly higher levels of TNF-α than untreated mice at the 30th day p.i., and 30 days after treatment interruption, these levels started falling, ending up at levels lower than those at the beginning of the treatment.

TGF-β is a constitutively expressed cytokine in the lungs and a fundamental mediator for healing and fibrotic processes. TGF-β had its levels reduced by 5 d.p.i., probably due to the inflammatory process initiated in the lungs after the BLM stimulus. The levels of this cytokine increased again and stayed high at the 60th d.p.i. Concomitantly, there was an increase in collagen accumulation and tissue remodeling, consistent with the presence of TGF-β, TNF-α, and other cytokines [10,14,15,44,47,48,49,53,54]. TGF-β is secreted in its inactive form in the tissue, and interaction with integrin αvβ6 is necessary to break its latency. Activation of TGF-β by ανβ6 integrin plays a central role in the development of PF, appearing increased both in human lungs with the usual interstitial pneumonia pattern as well as in mouse lungs after the BLM challenge [55]. This integrin is produced by epithelial cells, and the itgb6 gene regulates its β6 subunit. This increase is partly due to the rise in active TGF-β in lung tissue, which induces the expression of *itgb6*, as suggested by the PCR array results obtained in this work, creating an amplification cycle of central factors in the development of IPF [55]. TGF-β decreased in animals treated with ToAP3 and the accumulation of *itgb6* transcript, but not in animals treated with ToAP4. This difference may be related to differences in the mechanisms of action of each of these peptides to exert their antifibrotic effect through their regulation of IR. Furthermore, animals treated with ToAP3 had a drop in TGF-β production after treatment was discontinued, coinciding with the same time that IL-1β rose again.

The *Smad4* gene is also closely linked with TGF-β signaling and related to α-smooth muscle actin (α-SMA) production and fibroblast differentiation into myofibroblasts [56,57]. The SMAD4 protein is part of several signaling pathways, including TGFBR activation, which leads to increased transcription of αvβ6 integrin. However, *Smad4* appeared to be downregulated at the beginning of the fibrotic process in BALB/c mice. One possible explanation for that is that, as the PCR technique is used to verify transcript accumulation, the decrease in SMAD4 mRNA levels may not be directly related to its negative regulation but rather to its high translation. This is because *Smad4* participates in signaling pathways involved in the fibrotic process as well as in the path that leads to the production of α-SMA, which is quite active in myofibroblasts [56,57]. In addition, just as we found lower levels of transcript accumulation in this work, patients with carcinogenesis-associated IPF appear to have common *Smad4* expression in lung tissue compared to trachea cells or normal lung tissue [58]. Thus, the lower transcript accumulation of this gene may be associated with two opposite situations: lower transcription or even high transcription related to a higher translation rate, generating low levels of mRNA accumulation. Likewise, as the accumulation of *Smad4* transcripts decreased at the beginning of the fibrotic process, probably because of a high translation in that period, *Stat6*, which contributes to a wide variety of IL-4 and IL13-induced functions (IL-4Rα pathway), resulting in local lung responses such as airway hyperresponsiveness, goblet cell hyperplasia, and tissue remodeling, displayed low levels of transcript accumulation on the 15th d.p.i. [59,60,61]. The drop that was observed in STAT6 mRNA levels coincides with the peak of IL-13 production, indicating that the high recruitment of the protein may have led to an increase in translation and, consequently, to the decrease observed in the accumulation of the *Stat6* transcript. In addition, an increase was observed in the mRNA accumulation of MMP-13 and FasL, which are associated with the inflammatory process that occurs for the maintenance of fibrosis and disease severity [58,59,60,61,62,63,64].

Regulatory cytokines and cytokines with a Th2 profile play an essential role during the inflammatory and maintenance phases of IPF [2]. IL-10 is a regulatory cytokine, and its increased production is associated with a rise in the pro-inflammatory cytokine production that accompanies BLM lung injury. We observed increased production of IL-10 in the first 15 days. While untreated animals exhibit high levels of TNF-α and IL-1β, both associated with NF-κB, as well as IL-10, the IL-10 levels decline over time. The increase in IL-10, together with IL-4, plays a role in inducing the M2 profile of macrophages, which, in turn, plays an essential role in the healing process and in fibrotic scar formation after tissue injury [65]. The increasing levels of IL-10 that we observed in our model in the first 15 days could have contributed to the change in the profile of M1 macrophages to M2 and fibrocyte recruitment, which is needed at the outset of the collagen deposition process that we observed on the 15th day. When evaluating ToAP3- or ToAP4-treated mice, a significant decrease in the levels of IL-10 was observed in these animals, even 30 days after the last dose of the peptides. Furthermore, the Th2 profile cytokine we sought for changes in peptide-treated animals was IL-13. This is considered a profibrotic cytokine and can activate TGF-β, whose increase is associated with arginase increase and has been associated with the development of various chronic inflammatory diseases and fibrotic diseases [44,54,66,67]. IL-13 is detected in the bronchial lavage of IPF patients. Their fibroblasts are hyper-responsive to this cytokine, and its expression and receptor are correlated to the severity of the disease [68]. In this context, its role is more important than that of IL-4, as it can selectively induce the production of TGF-β [15,66]. In our assays, ToAP3- or ToAP4-treated animals had significantly lower IL-13 levels than animals after 30 days of BLM instillation. This difference remains 30 days after the end of treatment in the ToAP3-treated mice, while it disappears in the ToAP4-treated animals. As IL-13 is of great importance for fibrotic development in IPF, the drop in the levels of this cytokine is consistent with the interruption of the process caused by treatment with ToAP3 and ToAP4.

Gene expression in the pulmonary tissue of BLM-instilled mice treated with ToAP3 or ToAP4 contributed to the idea that, although the peptides have similar primary structures, they act differently to control the fibrotic process. Both peptides have been described as capable of interfering with the production of inflammatory mediators by interacting with membrane receptors [23]. These data corroborate the findings in this work, which show the ability of both peptides to regulate cytokines produced in the fibrotic context. However, the differences observed in the results of PCR array assays are striking, with ToAP4-treated animals showing little ability to alter mRNA accumulation in mice’s pulmonary tissue compared to untreated animals. The few alterations observed in these assays, added to the other data presented here, suggest that the regulation exerted by ToAP4 could be post-transcriptional for most genes or is associated with other cell signaling pathways that should be investigated.

Differently, ToAP3 seems able to regulate several factors associated with experimental IPF at the transcriptional level. The treatment with this peptide was able to alter the accumulation of Akt1, which is a protein highly expressed in the lungs of patients with IPF and which is related to cell survival by mediating growth, metabolism, and ROS production, as well as the regulation of TNF- α [63,64,69]. Furthermore, ToAP3 appears to have downregulated the expression of many profibrotic factors and maintained an inflammatory process, delaying fibrosis formation.

Chemokines such as CCL2, CCL6, CCL9 (in rodents), CCL17, CCL18, CXCL2, CXCL11, and their receptors have been reported to be related to fibrotic development, both associated with stimulation of fibrogenesis and its regulation [15,53,66,70]. Only Ccl11 and Ccr2 genes were modulated in our analysis. However, in the context of experimental IPF, both our peptides could inhibit the migration of cells to the lungs, as noted by histopathological analysis. This probably occurred through regulating the inflammatory process in the lungs, as evidenced by the regulation of cytokines found in the pulmonary tissue. In addition to TNF-a, another pro-inflammatory cytokine important in the IPF context is IL-1β. This cytokine can induce acute lung damage and contribute to PF progression [44,71]. Additionally, IL-1 can increase fibroblast production in a PDGF-dependent manner, increasing collagen production, glycosaminoglycan, and collagenase activity [45,72]. It has been reported that IL-1 detection in experimental models occurs only in the early days after BLM stimulation. Its role seems more relevant at the beginning of the process, leading to IPF [27,73]. Thus, IL-1β has a controversial role in the development of fibrosis. While this cytokine may be necessary during the beginning of the fibrotic process, the expression of high levels of it at a later stage could act as an inducer of myofibroblast apoptosis [7,27]. In the results shown in this work, it was possible to follow the production of IL-1β after BLM instillation in mice. We were able to observe increasing production of IL-1β, accompanying fibrosis development. In addition, the results seen in ToAP3 and ToAP4-treated animals showed IL-1β levels diminished by the time the treatment ended, and after 30 days, levels increased significantly.

Evaluating all the cytokines produced, a very high production of TNF-α, accompanied by high levels of NF-κB transcript, could, together with other factors, prevent the change from the inflammatory profile to a fibrotic profile and, thus, prevent the development of the disease. Interestingly, the high levels of TNF-α in the lung interstitium at the end of treatment with both peptides were not accompanied by high levels of IL-1β or IL-10, both associated with NF-κB.

The data generated during this work shows the immunomodulatory capacity of ToAP3 and ToAP4 peptides to reduce the establishment of PF. By different mechanisms, treatment with ToAP3 and treatment with ToAP4 could efficiently and lastingly control experimental IPF, preserving the animals’ tissue by regulating the inflammatory process resulting from BLM instillation. Thus, ToAP3 and ToAP4 can be considered therapeutic adjuvants and alternatives for preventing and treating IPF. However, further analysis is needed to understand how these peptides interact with the immune system to regulate fibrotic development.

## 4. Materials and Methods

### 4.1. Mice

BALB/c mice (*Mus musculus*), both male and female, aged 8 to 12 weeks (*n* = 50), were used. Each experimental group was composed of 5 animals, and the results represent three independent experiments. All animals were kept at the Institute of Biological Sciences animal facility at the University of Brasilia (UnB), with water and food ad libitum. All the procedures followed the requirements of the Animal Ethics Committee (CEUA), UnB (UnBDoC no. 100166/2014).

Animals were anesthetized (Ketamine and Xylazine, 80 and 10 mg·kg^−1^, respectively) and submitted to a single BLM intratracheal instillation with 50 µL of BLM at 5 U/mL and evaluated for 60 days, during which time the animals were weighed and the survival rate was monitored. They were submitted to all tests described below, and lung material was collected for histopathological evaluation, ELISA, and PCR tests.

### 4.2. Anti-Inflammatory Peptide Treatment

Peptides ToAP3 (FIGMIPGLIGGLISAIK-NH_2_) and ToAP4 (FFSLIPSLIGGLVSAIK-NH_2_) [29] were synthesized by FastBio (Ribeirão Preto, SP, Brazil) using the solid-phase F-MOC strategy (N-9-fluorophenyl methoxy-carbonyl). Peptide analyses were performed by MALDI-TOF reflection (MALDI Autoflex Speed TOF/TOF Bruker Daltronics, Billerica, MA, USA) operated in positive mode with positive reflection; the molecular mass was calculated using Isotope Pattern (version 3.4 Build 76, Bruker Daltonics) as described in Veloso et al., 2019 [23].

All concentrations chosen for treatment in this work were based on a previous study [23]. Five days after BLM instillation, mice were treated intranasally every three days with 20 µL of ToAP3 or ToAP4 in a concentration of 31.25 µM until the 30th day p.i. The treatment schedule and evaluations carried out were summarized in Figure 7.

### 4.3. Morphological Evaluation

For the histopathological evaluation, lung fragments from all groups were collected and kept in phosphate-buffered saline (PBS) with 10% formaldehyde until their inclusion in paraffin. Then, 5 µm slices of the material were fitted on glass slides, dyed with hematoxylin and eosin (HE) or Masson trichrome, and examined via light microscopy. All images were obtained using the Aperio CS2 (Leika Biosystems, Nussloch, Germany) slide scanner and ImageScope 12.3.3 software.

The collagen deposition was also analyzed by Col. I (Collagen I alpha 1, Novos Biologicals, Centennial, CO, USA), Col. III (Collagen III alpha 1/COL3A1, Novus Biologicals), and Col. IV (Collagen IV alpha 1, Novus Biologicals) using the immunohistochemistry (IHC) technique. All images were obtained using the Aperio CS2 (Leika Biosystems) slide scanner and ImageScope 12.3.3 software, and the collagen quantification on each slide was performed by the positive pixel counting algorithm in the software. The color (range of hues and saturation) and three intensity ranges (weak, positive, and strong) were specified. For the pixels that satisfy the color setting, the algorithm counts the number and intensity sum in each intensity range. Data are shown as the number of positive pixels per area.

Animal µCT analysis is an important tool that allows us to follow the development and establishment of fibrosis in the same animals, as the technique does not require animal euthanasia. Micro-computed tomography (µCT) images were acquired using the Albira CT system (PET/SPECT, Albira, Bruker, Billerica, MA, USA) [74]. Every five days p.i., mice were anesthetized and positioned on the ventral decubitus for image acquisition. Data were obtained using the factory-calibrated High Res Albira settings (1000 projections; 70 mm FOV; 45 KVp; and 400 mA). CT image reconstruction was made with the Albira reconstruction module using the standard option (Albira Software Suite, version 3.9, Bruker, Billerica, MA, USA). PMOD software (PMOD Technologies LLC, Zurich, Switzerland) version 3.9 was used to achieve whole pulmonary cavity quantification in Hounsfield Units (HU). The XRM-5 X-ray Phantom Mouse (caliper) was used to create standard parameters for lung density.

The density of lung tissue was analyzed by segmenting the aerial volume of the lungs and obtaining a histogram of the Hounsfield units (HU) and their corresponding voxel frequencies. The HU scale is a standard measurement for CT and X-ray analysis, representing the attenuation coefficients of various materials. This scale was established by measuring the attenuation coefficients of water and air, which are 0 and −1000, respectively. Therefore, air and water have radiodensities of −1000 and 0, respectively. HU values higher than 0 can also be calculated, which are commonly used to measure biological tissues. For example, bone tissue typically has a HU measurement of about 200–400 HU, while soft tissue such as the liver can measure around 60 HU.

The CT scanner produces 3D lung images by combining multiple projections. Each 2D projection consists of pixels, which are the smallest image units. In 3D images, voxels represent 3D pixels. To analyze the experimental lung data, we created a histogram that compared the frequency of voxels with their corresponding HU values for each mouse. This analysis allowed us to identify regions with different X-ray attenuation levels. Denser regions indicate the presence of solid masses, while less dense regions represent healthy aerial lung spaces. Fibrotic areas with higher HU values were later identified as solid masses in the histopathology analysis.

### 4.4. Pulmonary Ventilatory Mechanics

Animals were anesthetized and tracheostomized with an 18 gauge metal cannula. Afterward, mice were paralyzed with pancuronium bromide intraperitoneal injection (0.1 mg/kg), attached to a computer-controlled FlexiVent FX mechanical ventilator (Scireq, Montreal, QC, Canada), and mechanically ventilated in the volume-controlled mode in room air with a tidal volume of 8 mL/kg body weight and a respiratory frequency of 100 breaths/min against a positive end-expiratory pressure (PEEP) of 3 cmH_2_O. Respiratory system resistance (Rrs) and elastance (Ers) were measured by applying a volume signal perturbation containing one frequency and fitting volume, airflow, and airway opening pressure signals to the respiratory system equation of motion [75]. Respiratory system mechanical impedance (Zrs) was evaluated by applying volume signal perturbation containing frequencies between 0.25 and 19.62 Hz for the airway opening, and Zrs was fitted to the constant-phase model to obtain estimates of airway resistance (Rn), tissue damping (G), and tissue elastance (H) [76]. The parameters accepted the model fit if the coefficient of determination was >0.95. Ten measurements of Rrs, Ers, and Zrs were performed, and values were averaged.

### 4.5. PCR Array

The entire lung tissue of mice in the control and experimental groups, peptide-treated or untreated, was extracted using the RNeasy Mini Kit (Qiagen, Hilden, Germany), followed by treatment with DNAseI (Qiagen) to ensure that the final product is free from genomic DNA contamination, following the manufacturer’s instructions. After quantitative and qualitative evaluation of the sample, the RNA was reverse transcribed to complementary DNA (cDNA) using the RT2 First Strand Kit (SA Biosciences, Frederick, MD, USA) according to the protocol provided by the manufacturer. The samples were then mixed with RT^2^ *SYBR^®^ Green ROX qPCR Master Mix* (SA Qiagen, Venlo, The Netherlands) and added to the RT^2^ *Fibrosis PCR array*, 96 wells (PAMM-120Z RT^2^ Profile™ PCR Array Qiagen), as per the manufacturer’s instructions. The genes evaluated are described in Table 3.

### 4.6. Cytokine Quantification

Mice lungs were collected, weighted, and macerated with 1 mL of phosphate-buffered saline (PBS) (interstitium). The supernatant was collected after centrifugation at 300× *g* for 5 min and stored at −20 °C. Cytokines TNF-α, IL-1β, IL-10, IL-13, and TGF-β were quantified by an ELISA assay Ready-SET-Go!^®^ (eBioscience, San Diego, CA, USA) following the manufacturer’s instructions.

### 4.7. Statistical Analyses

Pulmonary ventilatory mechanics, percentage of VOIs, and cytokine quantification assays were analyzed by a two-way ANOVA followed by the Tukey post-test. Pulmonary density measures obtained in µCT analyses and lung transcript quantification were analyzed by one-way ANOVA followed by the Tukey post-test. Both results were expressed as the mean ± SEM of the representative data. Animal weight curves were the result of linear regression analyses.

All statistical analyses were performed using GraphPad Prism version 7.0. Data were considered significant when *p* < 0.05.

## Figures and Tables

**Figure 1 ijms-24-11049-f001:**
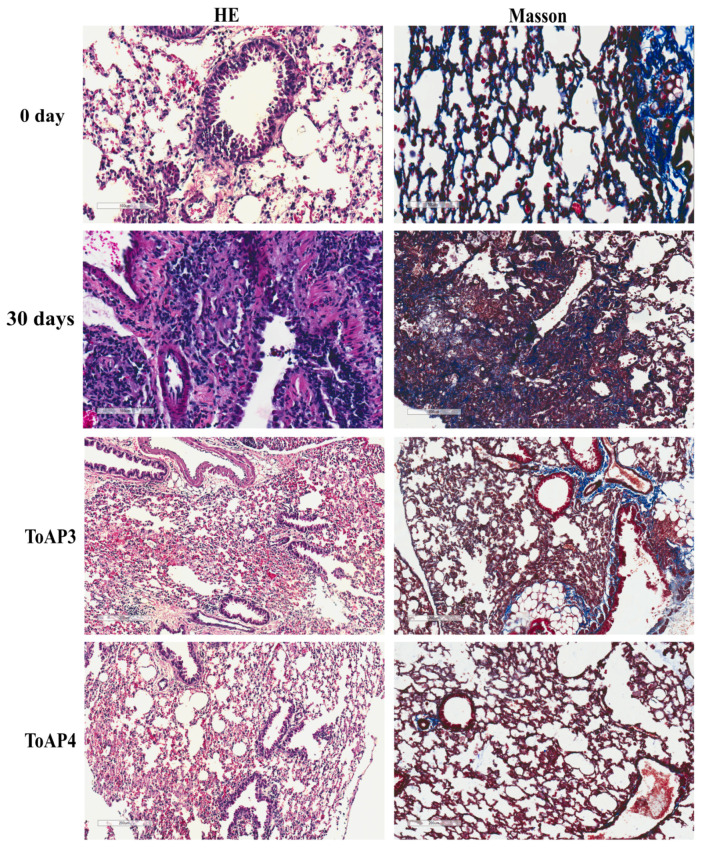
Morphological evaluation of BLM-instilled mice treated with ToAP3 or ToAP4: histopathological analysis. BALB/c mice were intratracheally instilled with BLM (50 µL, 5 U/mL), and on the 5th day, the intranasal treatment with each peptide started. Every 3 days, mice were treated intranasally with 20 µL (31.25 µM) of ToAP3 or ToAP4 and had pulmonary tissue collected and evaluated. Pulmonary tissue from peptide-treated mice and untreated mice (HE and Masson). 0 day and 30 days HE, magnification 200×; 30 days Masson, ToAP3 and ToAP4, magnification 100×. Results represent two independent experiments, each with five animals per experimental group.

**Figure 2 ijms-24-11049-f002:**
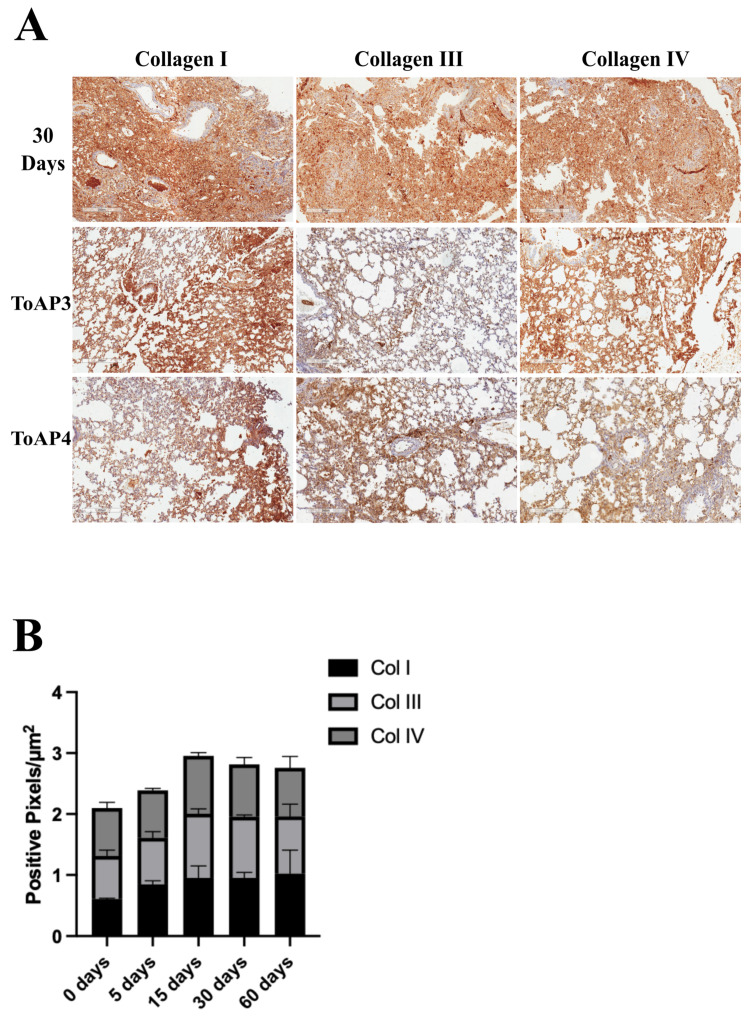
Morphological evaluation of BLM-instilled mice treated with ToAP3 or ToAP4: immunohistochemistry analysis. BALB/c mice were intratracheally instilled with BLM (50 µL, 5 U/mL), and on the 5th day, the intranasal treatment with each peptide started. Every 3 days, mice were treated intranasally with 20 µL (31.25 µM) of ToAP3 or ToAP4 and had pulmonary tissue collected and evaluated. Pulmonary tissue of peptide-treated mice and untreated mice with staining for Col. I, Col. III, and Col. IV. (**A**) Histological images of the experimental groups (magnification 100×) and (**B**) verification of their number of positive pixels per area. Results represent two independent experiments, each with five animals per experimental group.

**Figure 3 ijms-24-11049-f003:**
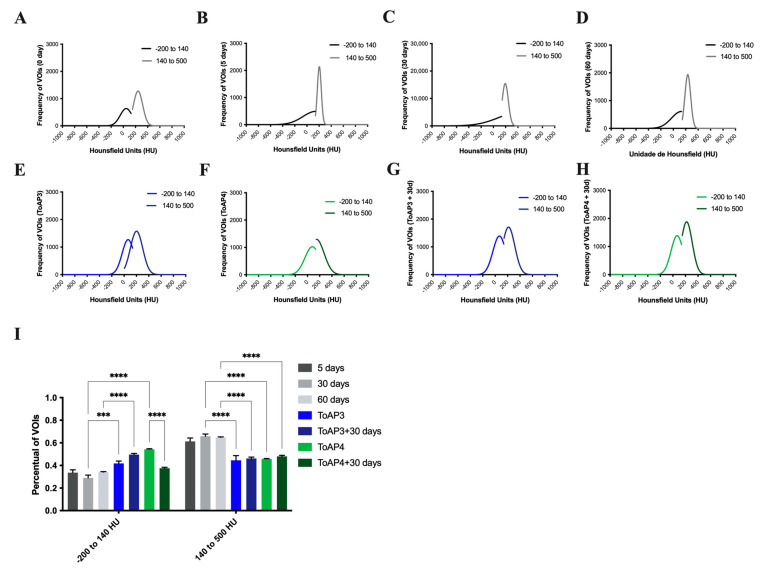
Morphological evaluation of BLM-instilled mice treated with ToAP3 or ToAP4: microCT analysis. BALB/c mice were intratracheally instilled with BLM (50 µL, 5 U/mL), and on the 5th day p.i., the intranasal treatment with each peptide started. Every 3 days, mice were treated intranasally with 20 µL (31.25 µM) of ToAP3 or ToAP4. The treatment was stopped on the 30th day p.i. We followed these animals for 30 more days. Mice were submitted for a µ-CT scan to evaluate their pulmonary density. We took the frequency of VOIs of these mice, the difference in the frequencies (**A**) at 0 days post-intillation (**B**) at 5 days post-instillation, (**C**) at 30 days p.i., (**D**) at 60 days p.i., (**E**) mice treated with ToAP3, (**F**) mice treated with ToAP4, (**G**) 30 days after stopping the treatment of mice with ToAP3, (**H**) 30 days after stopping the treatment of mice with ToAP4, and (**I**) the percentage of VOIs between −200 and 500 HU. Results represent two independent experiments, each with five animals per experimental group. *** *p* < 0.001, and **** *p* < 0.0001. Therefore, the treatment decreased the lung opacity observed during the analyzed time, and its suspension did not reverse the amelioration observed. It is important to note that while µCT analyses can detect differences in structural opacity compared to X-ray, they are unable to identify specific tissue components. Histological investigations are necessary to accurately determine the presence of tissular components. Hence, regarding the progression of fibrosis, we cannot confirm the typical deposition of collagen. However, both µCT and histopathological analysis reveal a distinct structural difference between the control and peptide-treated mice.

**Figure 4 ijms-24-11049-f004:**
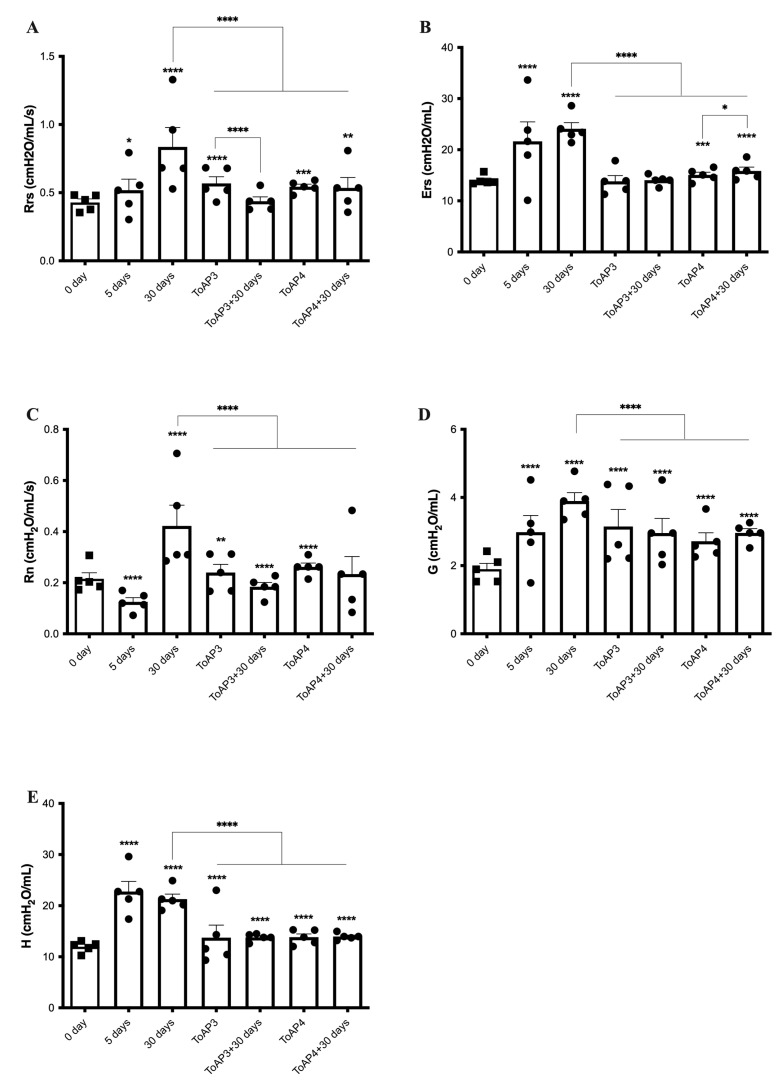
Evaluation of ventilatory mechanics in BML-instilled mice after treatment with ToAP3 or ToAP4. BALB/c mice were intratracheally instilled with BLM (50 µL, 5 U/mL), and on the 5th day p.i., the intranasal treatment with each peptide started. Every 3 days, mice were treated intranasally with 20 µL (31.25 µM) of ToAP3 or ToAP4. On the 5th or 30th day p.i., mice were sedated and anesthetized with ketamine and xylazine (15 mL/kg and 0.1 mL/kg, respectively) and submitted to tracheostomy to be connected to the mechanical ventilator. (**A**) Respiratory system resistance—Rrs; (**B**) respiratory system elastance—Ers; (**C**) Newtonian resistance—Rn; (**D**) tissue cushioning—G; and (**E**) tissue elastance—H. * *p* < 0.05, ** *p* < 0.01, *** *p* < 0.001, and **** *p* < 0.0001.

**Figure 5 ijms-24-11049-f005:**
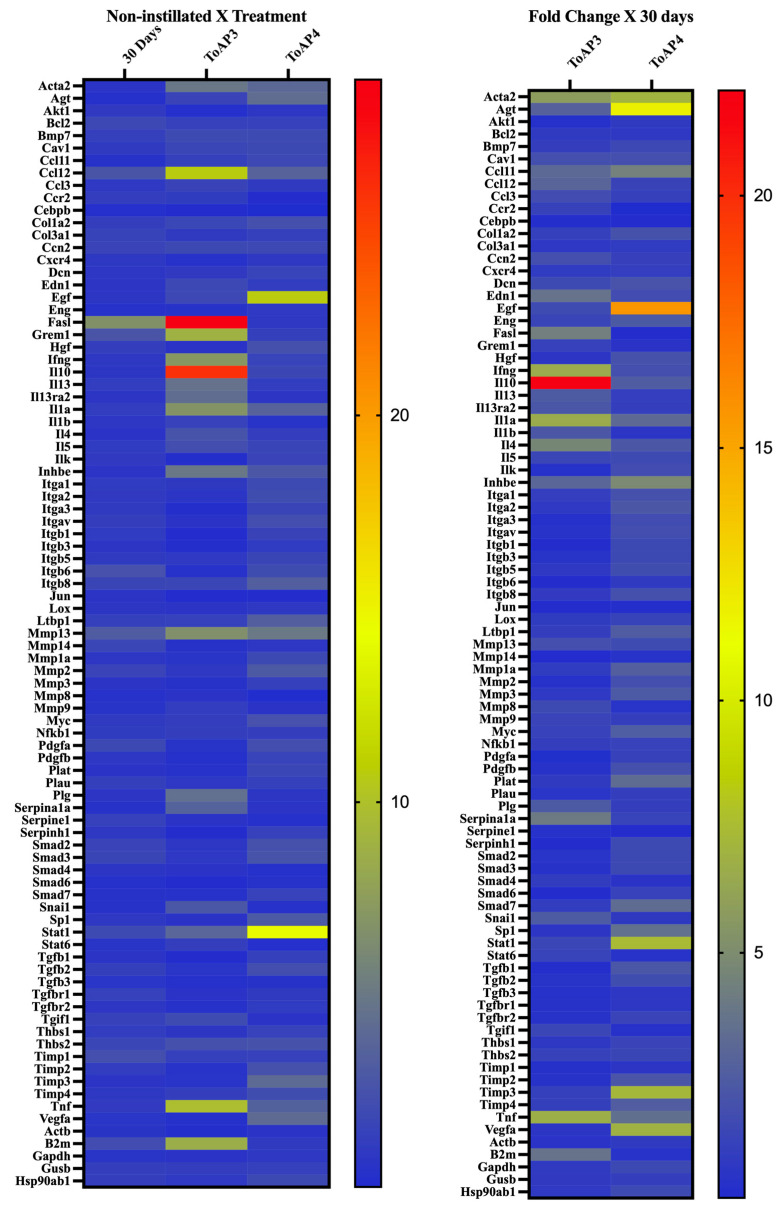
Heatmap of genes analyzed in a PCR array. BALB/c mice were intratracheally instilled with BLM (50 µL, 5 U/mL), and on the 5th day, the intranasal treatment with each peptide started. Every 3 days, mice were treated intranasally with 20 µL (31.25 µM) of ToAP3 or ToAP4. On the 30th day p.i., animals were euthanized and pulmonary tissue collected for mRNA extraction and PCR array analysis. The figure shows the fold change of all genes analyzed.

**Figure 6 ijms-24-11049-f006:**
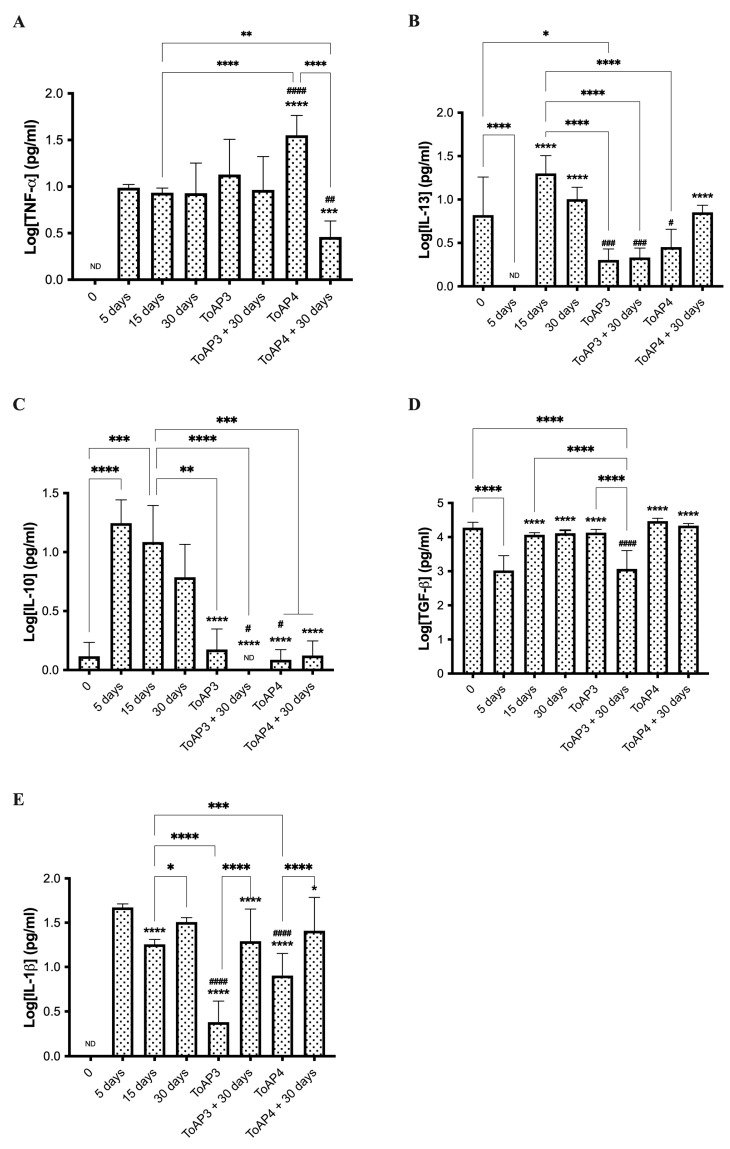
Concentration of cytokines found in the pulmonary interstitium of BLM-instilled mice treated with ToAP3 or ToAP4. BALB/c mice were intratracheally instilled with BLM (50 µL, 5 U/mL), and on the 5th day p.i., the intranasal treatment with each peptide started. Every 3 days, mice were treated intranasally with 20 µL (31.25 µM) of ToAP3 or ToAP4. On the 5th, 30th, or 60th day p.i., mice were euthanized, pulmonary tissue samples were collected and macerated, and the supernatant was saved for cytokine measurement. Quantification of levels of (**A**) TNF-α, (**B**) IL-13, (**C**) IL-10, (**D**) TGF-β, and (**E**) IL-1β. * *p* < 0.05, ** *p* < 0.01, *** *p* < 0.001, and **** *p* < 0.0001. # *p* < 0.05, ## *p* < 0.01, ### *p* < 0.001, and #### *p* < 0.0001, compared with 30 days. ND = non-detected.

**Figure 7 ijms-24-11049-f007:**
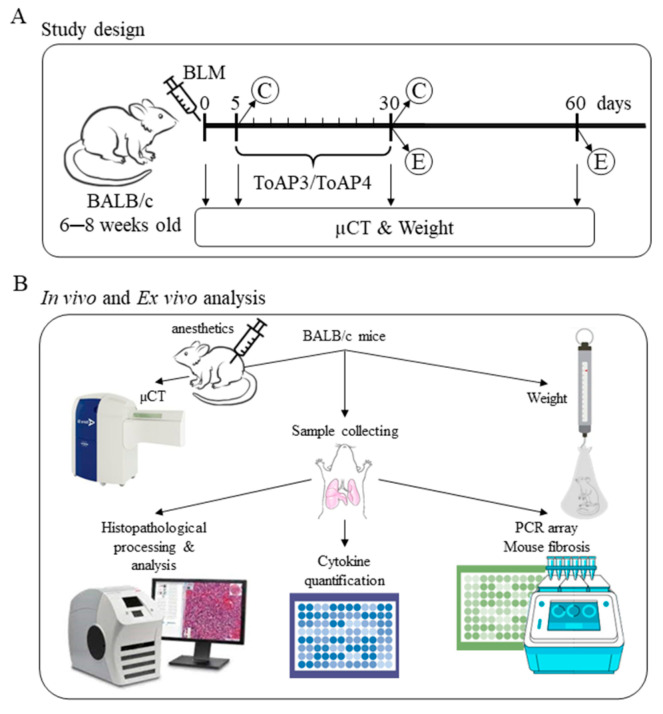
Workflow. (**A**) BALB/c mice were intratracheally instilled with BLM (50 µL, 5 U/mL), and on the 5th day p.i., the intranasal treatment with 20 µL (31.25 µM) of ToAP3 or ToAP4 started (C) and was maintained every 3 days until the 30th day p.i. A group of animals was then euthanized (E), and another group was supervised until the 60th day p.i. (**B**) Schematic design of the in vivo and ex vivo analyses of mice.

**Table 1 ijms-24-11049-t001:** Quantification of accumulation of transcript genes related to pulmonary fibrosis at 15 and 30 p.i., verified by PCR array, compared with non-instilled mice.

Symbol	Description	15th Day		30th Day	
		Fold Change	*p* Value	Fold Change	*p* Value
Eng	Endoglin	0.47	0.349698	0.37	0.038711
Fasl	Fas ligand (TNF superfamily, member 6)	0.65	0.352122	6.68	0.039064
Itgb6	Integrin beta 6	3.06	0.014495	2.44	0.019880
Smad4	Homolog of MAD 4 (Drosophila)	0.48	0.005721	0.62	0.027655
Stat6	Transducer and activator of transcription 6	0.30	0.033396	0.54	0.078273
Mmp13	Matrix metalloperoxidase 13	6.21	0.149880	3.09	0.029585

The color of the values in the table represents the fold change of the differentially expressed genes, with positive regulation in red and negative regulation in blue. The table shows comparisons between genes that presented fold change <0.5 or >2 and *p* ≤ 0.05.

**Table 2 ijms-24-11049-t002:** Quantification of transcript gene accumulation related to pulmonary fibrosis after 30 days of treatment, verified by PCR array, in comparison with mice on the 30th day post-instillation.

Symbol	Description	ToAP3 Treatment		ToAP4 Treatment	
		Fold Change	*p* Value	Fold Change	*p* Value
*Akt1*	Proto-oncogene Viral de Timoma 1	0.34	0.043212	0.83	0.570278
Ccl11	Chemokine (motivo c-c) ligante 11	3.18	0.021522	4.41	0.323722
Ccr2	Chemokine RECEPTOR (motive c-c) 2	1.18	0.593361	0.14	0.043364
Fasl	Fas ligand (TNF superfamily, member 6)	4.30	0.006065	0.20	0.060712
Il1a	Interleukin 1 alpha	6.53	0.000911	3.19	0.239425
Il1b	Interleukin 1 beta	2.27	0.046645	0.62	0.750740
Itgav	Integrin alpha V	0.46	0.045766	1.79	0.375390
Itgb6	Integrin beta 6	0.21	0.005560	0.83	0.678812
Mmp14	Matrix metallopeptidase 14	0.18	0.015702	0.46	0.278868
Mmp2	Matrix metallopeptidase 2	0.39	0.043679	1.86	0.365113
Pdgfa	Platelet-derived growth factor, alpha	0.25	0.020553	1.18	0.536549
Serpine1	Serine (or cysteine) peptidase inhibitor, clade E, member 1	0.40	0.026483	0.24	0.000152
Serpinh1	Serine (or cysteine) peptidase inhibitor, clade H, member 1	0.21	0.025890	1.59	0.358584
Smad4	SMAD family member 4	0.92	0.778855	0.50	0.035041
Tgfb1	Transforming growth factor, beta 1	0.28	0.029165	2.23	0.338805
Tgfb2	Transforming growth factor, beta 2	0.47	0.019035	1.74	0.355887
Tgfb3	Transforming growth factor, beta 3	0.37	0.011840	0.71	0.155044
Tgfbr1	Transforming growth factor, beta receptor I	0.44	0.035557	0.71	0.142083
Tgfbr2	Transforming growth factor, beta receptor 2	0.45	0.000265	1.34	0.038078
Smad6	SMAD family member 6	0.22	0.025221	1.20	0.562480
Timp2	Tissue inhibitor of metalloproteinase 2	0.45	0.003830	2.19	0.310287

The color of the values in the table represents the fold change of the differentially expressed genes, with positive regulation in red and negative regulation in blue. The table shows comparisons between genes that presented fold changes <0.5 or >2 and *p* ≤ 0.05.

**Table 3 ijms-24-11049-t003:** Genes analyzed in RT-PCR Array.

Genes’ Group	Name of Genes
Profibrotic	Acta2 (α-SMA), Agt, Ccl11 (Eotaxin), Ccl12 (MCP-5, Scya12), Ccl3 (Mip-1a), Ccn2, Grem1, Il13, Il13ra2, Il4, Il5, Snai1 (Snail).
Antifibrotic	Bmp7, Hgf, Ifng, Il10, Il13ra2
Extracellular Matrix (ECM) & Cell Adhesion Molecules	Col1a2, Col3a1. Mmp13, Mmp14, Mmp1a, Mmp2, Mmp3, Mmp8, Mmp9, Plat (tPA), Plau(uPA), PlgSerpina1a, Serpine1 (Pai-1), Serpinh1 (Hsp47), Timp1, Timp2, Timp3, Timp4 Itga1, Itga2, Itga3, Itgav, Itgb1, Itgb3, Itgb5, Itgb6, Itgb8
Inflammatory Cytokines & Chemokines	Ccl11 (Eotaxin), Ccl12 (MCP-5, Scya12), Ccl3 (Mip1a), Ccr2, Cxcr4, Ifng, Il10, Il13, Il13ra2, Il1a, Il1b, Il4, Il5, Ilk, Tnf
Growth Factors	Agt, Ccn2, Edn1, Egf, Hgf, Pdgfa, Pdgfb, Vegfa
Signal Transduction	TGFβ Superfamily Members-Bmp7, Cav1, Dcn, Eng (Evi1), Grem1, Inhbe, Ltbp1, Smad2, Smad3, Smad4, Smad6, Smad7, Tgfb1, Tgfb2, Tgfb3, Tgfbr1 (ALK5), Tgfbr2, Tgif1, Thbs1 (TSP-1), Thbs2. Transcription Factor-Cebpb, Jun, Myc, Nfkb1, Sp1, Stat1, Stat6
Epithelial-to-Mesenchymal Transition (EMT)	Akt1, Bmp7, Col1a2, Col3a1, Itgav, Itgb1, Mmp2, Mmp3, Mmp9, Serpine1 (Pai-1), Smad2, Snai1 (Snail), Tgfb1, Tgfb2, Tgfb3, Timp1
Other Fibrosis Genes	Bcl2, Fasl

## Data Availability

Not applicable.

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
