# Peer review of "Innovative Pre-Clinical Data Using Peptides to Intervene in the Evolution of Pulmonary Fibrosis"

_ijms, 2023, doi:10.3390/ijms241311049_

Round 1
Reviewer 1 Report
The authors have demonstrated that the immune modulatory peptides, ToAP3 and ToAP4 ameliorate the bleomycin-induced long-lasting lung fibrosis in BALB/c mice. In this point, this work is new and informative. However, there seems to be several errors and over-speculation in this manuscript. Therefore, the authors should reply to the following queries:
[Major points]
1) How was the pathological profile of entire lobe? Judging from lesioned sites at 30 days post instillation (Fig.1 and Suppl. Fig. 1), severe fibrosis in the lung was progressed. Nevertheless, both body weight loss and survival rate were rarely affected. Tiling profile of section picture at low-power field may partly help these phenomena.
2) Negative controls (non-instillated / 0 day) are missing in several figures (Figs. 2, 3 & Suppl. Fig. 4). The authors should put each data appropriately and revise figures, which let the readers understand this manuscript better.
3) ToAP3 as well as ToAP4 clearly ameliorate the BLM-induced lung fibrosis with similar potency (Figs 1&4). However, the differentially expressed genes in the lung treated with ToAP3 and ToAP4 showed a quite different pattern between the two peptides. The authors state the possibility that ToAP4 may regulate the genes at the posttranscriptional level. What is the reason for your speculation? Likewise, if the inhibitory mechanism could be different between the two peptides, concomitant application of the two peptides would show more successful therapeutic effects on the BLM-induced lung fibrosis. The authors should elucidate this possibility. By the way, which cell is affected by each peptide?
4) Resolving established fibrosis rather than anti-inflammation strategy is currently thought to have great potential as a beneficial intervention in patients with IPF. The previous findings that TLR4 contributes to both fibroblast activation and progression of tissue fibrosis tempt us to think that ToAP3/ToAP4 may exert anti-fibrotic action partly through dedifferentiating myofibroblasts. How are the therapeutic effects if the treatment with ToAP3/ToAP4 were started at 15 dpi? The authors have evaluated this possibility?
[Minor points]
1) Please replace “IPF” with “PF” (on lines 118 & 135).
2) Please give appropriate abbreviations for “tissue damping” and “tissue elastance” (on lines 126 & 128). Tentative names, (G) or (H) are not suitable.
3) Please replace the HE-staining profile at 30 dpi with better one (Fig. 1).
4) Please give a detailed explanation for ISP because it is difficult for the readers to understand the quantitative analysis in Fig. 2 compared with that in Supplementary Fig. 4. The value of vertical axis is not log ISP but ISP (Fig. 2)? Is it correct?
5) -200 HU e 500HU -200 HU to 500HU (line 182)
6) The word “de” in the vertical axis of Fig. 3H “of”
7) The word “BML” in the subtitle of Fig.4 “BLM”
8) 0 days 0 day in Fig. 4
9) Please check and correct the corresponding Figs in parentheses on lines 294-300.
I will recommend English-language editing to the authors.
Author Response
Thank you for your comments. We have addressed the concerns raised by your comments. The text has been reviewed and corrected accordingly, substantially improving the manuscript. Suggested information and data have been added to the text, such as the reorganized discussion. We have answered your comments point-by-point.
1) Answer: We do not have the histopathological analysis of the entire lobe since the pathologist cut the tissue to be imbibed into paraffin. However, we added here the whole picture of the tissue fragment. In the images below, it is possible to observe the patterns of collagen deposition and inflammatory infiltration. For the histopathological model's description, the hole BLM instillation mice’s lung was extracted and fixed with a solution of 10% of formaldehyde and then processed to finally obtain slides with sequential cuts to find the pattern of the lung lesions. The description of the histopathological exam of the mice’s pulmonary tissue 30 d.p.i. described the alveolar spaces began to be replaced by conjunctive stroma, with fibroblasts, histiocytes, and lymphocytes. It was also possible to verify the existence of airways of different calibers represented by structures with walls constituted by fibromuscular and internally lined by respiratory pattern ciliated epithelium. Using the Masson trichrome stain, detecting the collagen in the connective stroma that was even thicker around the airways than in the prior time points was possible.
(The images are in the file attached)
2) Answer: This observation is accurate. We included the non-instillated data with the murine model kinetic analysis plotted in the supplementary material. The 0-day picture of Figure 2 is now in Supplementary Figure 3; the 0-day of Figure 3 was added, and the 0-day picture of mCT is in Supplementary Figure 4. And we added this information to the results section.
3) Answer: Our first point to study was managing the immune response activation and ToAP3 and ToAP4 ameliorate the BLM-induced lung fibrosis. The gene transcript accumulation in the lungs of animals treated with ToAP3 and ToAP4 showed a different pattern. We could observe that ToAP3 could modify the transcript accumulation of many genes related to the fibrotic process. At the same time, ToAP4, although it had a similar structure to ToAP3, and effects on tissue and cytokine levels, did not have the same impact on gene modulation. This and unpublished results led us to speculate that ToAP4 could exert its effect at the posttranscriptional level, which was better explained in the revised discussion.
We carried out experiments in vitro with both peptides, and both decreased TNF production (Veloso Junior et al., 2019). This article’s figure below showed that the peptides decreased TNF and IL-1b cytokines when the macrophages (A, C, E) and dendritic cells (B, D, F) were stimulated with LPS.
(The images are in the file attached)
Besides, in our unpublished data on sepsis induced by cecal ligation and puncture, we demonstrated that the levels of serum IL-6 were decreased after intraperitoneal treatment with ToAP3. We also showed that the survival curve is better with the treatment, and the count of cells in BAL is lower than in the non-treated mice. So, based on our previous data, we assumed that in vivo model of PF, the peptides are reducing the inflammatory response, allowing the mice to work out the injury better. In the discussion, we explained better the action of peptides based on our published data.
(The images are in the file attached)
So, we agree that as the inhibitory mechanism seems to be different between the two peptides, the concomitant application of the two peptides could show more success. We conducted some experiments to analyze if ToAP3 and ToAP4 can act synergistically to decrease cytokine production in vitro; however, they only showed addictive function on macrophages. Still, new experiments to test whether the peptides would have synergistic or summative activity can be designed with different cell types found in the context of pulmonary fibrosis to determine whether the effects achieved are the desired ones and which concentrations we should work with before new in vivo tests. We only finished exploring the peptide's effects on immune system cells (Veloso, 2019).
4) Answer: As described in the manuscript, protocols and therapeutic alternatives for IPF have been developed over the years but with little success in improving the quality of life or curing this disease. Despite the importance of cytokines in the regulation of the inflammatory process and their influence in experimental models of IPF, most therapeutic protocols using immunosuppressants fail in preclinical tests (Belperio et al., 2002; Borie, 2016; Sköld et al. al., 2017), speculating that the development of IPF would not be related to immune response, and it is only responsible for the chronic fibrotic process (Desai et al., 2018). Thus, considering that this happens due to the deregulation of the mechanisms responsible for tissue healing and that this usually occurs after tissue injury generated during the local inflammatory process, the use of elements that can regulate at the beginning of the fibrotic process can be interesting to understand the disease’s pathogenesis and to develop new therapeutical approaches. Therefore, we started the treatment with ToAP3/ToAP4 on the 5th-day p.i., where we could observe tissue alterations compatible with the inflammatory process without collagen deposition. The other point of interest is the possibility of using the peptides as an adjuvant to various disease treatments whose outcome is fibrosis formation, preventing tissue damage. Of course, established fibrosis is one important point to focus on, but our results presented here is the first part of the current project, and we do not have the answer about the late treatment until now.
[Minor points]
- Please replace “IPF” with “PF” (on lines 118 & 135).
Answer: We replaced it as indicated and checked the abbreviations.
- Please give appropriate abbreviations for “tissue damping” and “tissue elastance” (on lines 126 & 128). Tentative names, (G) or (H) are not suitable.
Answer: We thank you for your comment and concern regarding the appropriate abbreviations of tissue damping and tissue elastane, G, and H, respectively. Although it may sound unusual to readers interested in our paper, G and H were the abbreviations used by Hantos et al., the authors who described the foundations of the constant-phase model we employed to assess respiratory mechanics. Therefore, to maintain consistency with the spelling of these respiratory mechanics parameters, we have decided to retain the abbreviations G and H. Likewise, the abbreviations for “tissue damping” and “tissue elastance” (G) or (H) have been used in literature over the years. Some references are described below.
- Devos FC, Maaske A, Robichaud A, Pollaris L, Seys S, Lopez CA, et al. Forced expiration measurements in mouse models of obstructive and restrictive lung diseases. Respir Res. 2017;18(1):1–14.
- Smith BJ, Roy GS, Cleveland A, Mattson C, Okamura K, Charlebois CM, et al. Three Alveolar Phenotypes Govern Lung Function in Murine Ventilator-Induced Lung Injury. Front Physiol. 2020;11(June):1–14.
- Reiss LK, Raffetseder U, Gibbert L, Drescher HK, Streetz KL, Schwarz A, et al. Reevaluation of Lung Injury in TNF-Induced Shock: The Role of the Acid Sphingomyelinase. Mediators Inflamm. 2020;2020(i).
- Foong RE, Shaw NC, Berry LJ, Hart PH, Gorman S, Zosky GR. Vitamin D deficiency causes airway hyperresponsiveness, increases airway smooth muscle mass, and reduces TGF-β expression in the lungs of female BALB/c mice. Physiol Rep. 2014;2(3).
- Bozanich EM, Collins RA, Thamrin C, Hantos Z, Sly PD. Developmental changes in airway and tissue mechanics in mice. 2005;(January 2014).
- Doras C, Guen M Le, Peták F, Habre W. Cardiorespiratory effects of recruitment maneuvers and positive end expiratory pressure in an experimental context of acute lung injury and pulmonary hypertension. BMC Pulm Med. 2015;(April 2016).
- Larcombe AN, Foong RE, Berry LJ, Zosky GR, Sly PD. In utero cigarette smoke exposure impairs somatic and lung growth in BALB/c mice. 2011;38(4):932–8.
- Please replace the HE-staining profile at 30 dpi with better one (Fig. 1).
Answered: The figure was modified.
4) Please give a detailed explanation for ISP because it is difficult for the readers to understand the quantitative analysis in Fig. 2 compared with that in Supplementary Fig. 4. The value of vertical axis is not log ISP but ISP (Fig. 2)? Is it correct?
Answer: No, both axes were Log ISP. However, we decided to modify both graphs to represent better the changes we found in the tissue. A new analysis normalizing the number of positive pixels per area is now described in the figure. Also, the MM is now:
All images were obtained using Aperio CS2 (Leika Biosystems) slide scanner and the ImageScope 12.3.3 software, and the collagen quantification of the amount of the collagen marked in each slide was made by the positive pixel counting algorithm in the software. The color (range of hues and saturation) and three intensity ranges (weak, positive, and strong) were specified. For the pixels which satisfy the color setting, the algorithm counts the number and intensity‐sum in each intensity range. Data are shown in the number of positive pixels per area.
- -200 HU e 500HU à -200 HU to 500HU (line 182)
Answer: It was corrected.
- The word “de” in the vertical axis of Fig. 3H à “of”
Answer: It was corrected.
- The word “BML” in the subtitle of Fig.4 à “BLM”
Answer: It was corrected.
- 0 days à 0 day in Fig. 4
Answer: It was corrected.
- Please check and correct the corresponding Figs in parentheses on lines 294-300.
Answer: All the figures in parentheses described in the text were corrected. The mention was to cytokine production, which is in Figure 6.
I will recommend English-language editing to the authors.
Answer: The text was re-edited by a native English speaker.

Reviewer 2 Report
Idiopathic pulmonary fibrosis (IPF) is a form of chronic, progressive lung inflammation. Nintedanib and pirfenidone are drugs with antifibrotic properties that slow down the progression of the disease; however, the beneficial effect of treatment is achieved only in some patients.
To begin with, upon first reading the manuscript, I noticed the work done by the team for data collection and for the elaboration of this manuscript. The presented data are of high quality and convincing, as well as clinically very relevant. The manuscript is very well-written, properly documented.
I think the current version of this manuscript should be enriched to generate more interest from readers. I would kindly request the authors to include the following information in the introduction section (for example, in the form of a table), that they will describe
1) the most important scientific studies and research papers related to the current treatment of IPF (several), including completed clinical trials and ongoing research.
2) A diagram suggesting mechanisms leading to the fibrotic process in IPF.

Author Response
Thank you for your comments. We have addressed the concerns raised by your comments and we have answered your suggestions.
- the most important scientific studies and research papers related to the current treatment of IPF (several), including completed clinical trials and ongoing research.
Answer: In the discussion, we included some clinical trials of phase III in the course.
2) A diagram suggesting mechanisms leading to the fibrotic process in IPF.
Answer: We decided not to include this diagram because the exact block mechanisms of the peptides are not known, and other experiments are necessary to be done. Only the fibrosis process diagram will not add new information to the article. However, we better discuss the stages of the fibrotic process in the manuscript.
Reviewer 3 Report
In this study, the authors aim to evaluate the effects of peptides ToAP3 and ToAP4, in a lung fibrosis model induced in BALB/c mice by bleomycin instillation. Fibrotic progression was assessed through qualitative morphologic evaluation of H/E stained lung tissues, expression of collagens (I, III and IV), lung density by μCT analysis and function impairment through evaluation of pulmonary ventilatory mechanics (Rrs, Ers, and Zrs etc.). The authors also performed gene expression analysis and cytokine contents.
In my opinion, although the work could be of interest, requires major revisions before being published in International Journal of Molecular Sciences. Please see below my comments to the authors.
Major comments
· The authors used the BALB/c mouse strain considered more resistant to the induction of fibrosis by bleomycin than the C57BL6 strain, with the intention of developing a chronic PF model (see line 93: “We carried out the chronic PF model using BALB/c mice for this work”). Thus, it is important for the authors to demonstrate the development and maintenance of pulmonary fibrosis over a long period of time.
In this regard, the authors entitled the first paragraph of the Results section “The animal model strategy showed chronic inflammation with lung fibrosis formation”. However, they do not report any data demonstrating chronic inflammation. The only quantitative data are related to collagens expression (with a progressive increase of collagens I and III and instead a progressive decline of collagen IV except at day 60), and data related to respiratory mechanics, (with progressive decline except at day 60 when almost all parameters assume values similar to the control values (without treatment with bleomycin).
Moreover, data related to VOI’s frequency from micro-CT analysis (Fig 3) show again that at 60 days post-bleomycin, lung damage is reverted with respect to that observed at 30 days post-bleomycin, and that no progressive changes in VOI% were observed from 5, 30 to 60 days. Furthermore, the absence of control values collected at 60 days for ventilatory mechanics (Fig 4) and cytokines (Fig 6) prevents the reader from obtaining information about the long-term progression of these parameters.
Thus, the authors do not convincingly demonstrate the maintenance of fibrosis up to 60 days post-bleomycin.
As a consequence, all data demonstrating an improvement in parameters collected after treatments at 60 days post-bleomycin (groups ToAP3+30days and ToAP4+30days) can be attributed to the spontaneous reversal of fibrosis at this time. The authors have to seriously address this point, I think that they cannot consider the applied one, a chronic PF model.
· The authors should describe more realistically the data obtained.
For example, in the paragraph “The treatment decreases collagen deposition in lung tissue”, the title and the text do not match with the data shown in Figure 2. No treatment indeed exerts a significant effect, a trend towards a reduction of collagen expression is observed only with ToAP4. In addition, has to be noted that, in Figure 2, increased levels of collagen IV upon treatment with ToAP3, are interpreted by the authors as an “improvement of basal structural maintenance”, what this means in the context of lung fibrosis? And how do the authors interpret the collagen III growth trend observed after treatment with ToAP3?
Again, the authors should clearly indicate that treatments (see groups ToAP3+30days and ToAP4+30days) did not change VOI’s frequency and VOI% (-200 to 140 HU) with respect to the related control group (60 days).
· To allow the reader to compare the data reported for treated groups (ToAP3+30days and ToAP4+30days) with the related control group (60 days), the authors have to add in Figure 4 (Evaluation of ventilatory mechanics) and in Figure 6 (Concentration of cytokines….) the control values recorded at 60 days.
· The statistical significances in the graphs are unclear. I suggest their insertion only to allow the comparison between treated groups and related controls (group 30 days vs ToAP3 and vs ToAP4; and group 60 days vs ToAP3+30days and ToAP4+30days). Furthermore, the authors should add in the figure legends vs which experimental group the reported statistical significance is calculated.
· The discussion is too long and speculative. The authors should report only the most important data and above all emphasize the originality of their study compared to other studies
· In the discussion section, the authors claim statements that are not supported by the data reported in the study:
- Lines 329-330: “Indeed, in our experiments, μCT analysis showed an increasing opacity in the lungs of BALB/c mice during the 60 days following BLM instillation (see major comments reported above).
- Lines 335-336: “histopathological analysis corroborate the opacity in μCT images and the percentage of VOIs found in these exams, but they also suggest that although fibrosis persists, the inflammatory process decreases” What data do the authors refer to indicating that inflammation decreases?
- Lines 341-342: “… indicate that both peptides regulate inflammation in a manner consistent with their characteristics described in the literature and, consequently, in the development of fibrosis” Again what are the data indicating that peptides regulate inflammation? And again, see line 356 “It was assumed that the regulation of this inflammation by the peptides...”
- Lines 358-359: “Morphological analysis and ventilatory mechanics confirmed our hypothesis that immunomodulatory elements would regulate the inflammatory and scaring process.” What data do the authors refer from morphologic analysis indicating the immunomodulatory action of the peptides? The authors do not investigate the recruitment of inflammatory cells in the lung tissues, nor their activation, nor their inflammatory phenotype (for example M1 vs M2 macrophages…). The only data reported in this study associated with an inflammatory condition are the tissue levels of inflammatory cytokines.
- lines (line 390) “Together, these results indicate that ToAP3 and ToAP4 efficiently controlled the disease's progression, limiting cell migration to the pulmonary tissue.” Regarding the migration of immune cells, considering that the sentence is not supported by any data present in the paper, I would suggest either removing them or adding specific references.
- The manuscript requires substantial English revision. I suggest a language revision by a native English reviewer before resubmitting. Even in the tile, some inaccuracies are present (the use of the accent on the "é" in "Pré-clinical" is uncommon in scientific terminology, furthermore apostrophe after "fibrosis” seems unnecessary)
Minor comments
· Pag 2 line 48. The authors wrote: “Pulmonary injury induces cell migration”, I suggest to change in “Pulmonary injury induces immune cell migration”.
· How did the authors determine the dosage of peptides for the treatment? The authors should report the reasons for the choice in the text.
· In the paragraph “Cytokine quantification” the author should explain what they mean by and how they prepare the macerated lungs (interstitium). Furthermore, they mention the collection of bronchus alveolar lavage fluid (BALF), but no data are reported in this regard.
· In Tables 1 and 2 it has been reported “…. fold change >0.5 or >2 and p≤0.05”. It has to be changed in “…. fold change <0.5 or >2 and p≤0.05”.
· Authors cite non-English language papers; I recommend replacing them with papers written in English (see ref 23)
· The authors should cite more recent references
· Formatting is uneven (lines 174,175,176; sections 4.4, 4.5, 4.6, 4.7)
The manuscript requires substantial English revision. I suggest a language revision by a native English reviewer before resubmitting. Even in the tile, some inaccuracies are present (the use of the accent on the "é" in "Pré-clinical" is uncommon in scientific terminology, furthermore apostrophe after "fibrosis” seems unnecessary)
Author Response
Thank you for your comments. We have addressed the concerns raised by your comments. The text has been reviewed and corrected accordingly, substantially improving the manuscript. Suggested information and data were added to the manuscript when possible, such as the reorganized discussion. We have answered your comments point-by-point. The figures described in the answers are in the attached file.
1- Answer: The animal model using C57BL/6 mice instillated with BLM was not suitable for our project due to the high susceptibility of these animals to lung damage after IT treatment (Tashiro et al., 2017), with an exuberant inflammatory process, high mortalities in the first 15 days after treatment, and those that resist this period evolve to spontaneous cure, which makes it difficult to assess the effectiveness of drugs in preclinical tests (Agostini & Gurrieri, 2006; Tashiro et al., 2017; Ruscitti et al., 2018). This model fails to recapitulate several important features of unusual interstitial pneumonia (UIP), such as fibroblastic foci, hyperplastic epithelium, temporal heterogeneity, and honeycombing. Furthermore, after a single dose of BLM, an inflammatory process rich in neutrophils is observed in the C57Bl/6 model, more indicative of acute or Th17-mediated lung damage than in a fibrosis model (Scotton & Chambers, 2010; Degryse & Lawson, 2011).
It is important to underline that chronic inflammation represents a long-term reaction to an inflammatory stimulus characterized by continued recruitment of mononuclear leukocytes and tissue injury due to the sustained inflammatory response, lasting from weeks to a lifetime. Chronic inflammation is characterized by infiltration of mononuclear cells, tissue destruction, and tissue repair, involving angiogenesis and fibrosis (H.B. Fleit, 2014; Pahwa R et al., 2023). Macrophages are the dominant cells that are crucial cellular elements of chronic inflammation and are the primary source of cytokines such as TNF-α. Classical and alternative activated macrophages (M1 and M2 macrophages) regulate the chronic inflammatory outcome and have cytokine signatures IL-6, TNF-α, and TGF-β (Liu et al., 2021). Due to these characteristics of a chronic inflammatory response and the results found in our work on the cellular infiltrate in pulmonary tissue after 30 days of BLM instillation, cytokine profile, tissue remodeling, and gene expression, our model can be considered a chronic model of pulmonary inflammation, with fibrosis formation since the results show the same characteristics of the chronic inflammatory process. Although we changed the first section's title to “The animal model strategy showed long-term inflammation with lung fibrosis formation” to not induce a model misunderstanding by the readers and changed the chronic at line 93, as mentioned before.
Considering all the points described until now, we are demonstrating chronic lung inflammation. The histological data (Supplementary Figure 2 – 30- and 60-days post-instillation), the mCT images (supplementary figure 4 – white head arrows of 60 days post-instillation), and VOI analyses (Figure 3H) showed parameters that suggest worsening healing of the tissue and fibrosis formation. The ventilatory analysis returned to levels like the non-instilled mice because the mononuclear infiltration and edema decreased after lung healing; however, with fibrosis formation. New experiments might be done to confirm whether fibrosis will increase in this model after 60 dpi. As in the C57Bl/6 mice model, the BALB/c model shows limitations but gives us parameters to confirm our hypothesis about the peptides.
- Tashiro J, Rubio GA, Limper AH, Williams K, Elliot SJ, Ninou I, et al. Exploring Animal Models That Resemble Idiopathic Pulmonary Fibrosis. Front Med. 2017;4(July):1–11.
- Agostini C, Gurrieri C. Chemokine/cytokine cocktail in idiopathic pulmonary fibrosis. Proc Am Thorac Soc. 2006;3(4):357–63.
- Ruscitti F, Ravanetti F, Donofrio G, Ridwan Y, van Heijningen P, Essers J, et al. A multimodal imaging approach based on micro-CT and fluorescence molecular tomography for longitudinal assessment of bleomycin-induced lung fibrosis in mice. J Vis Exp. 2018;2018(134):1–7.
- Scotton CJ, Chambers RC. Bleomycin revisited: Towards a more representative model of IPF? Am J Physiol - Lung Cell Mol Physiol. 2010;299(4):439–41.
- Degryse AL, Lawson WE. Progress toward improving animal models for idiopathic pulmonary fibrosis. Am J Med Sci. 2011;341(6):444–9.
- B. Fleit. Chronic Inflammation. Editor(s): Linda M. McManus, Richard N. Mitchell. Pathobiology of Human Disease. Academic Press. 2014. Pages 300-314. ISBN 9780123864574. https://doi.org/10.1016/B978-0-12-386456-7.01808-6.
- Pahwa R, Goyal A, Jialal I. Chronic Inflammation. [Updated 2022 Aug 8]. In: StatPearls [Internet]. Treasure Island (FL): StatPearls Publishing; 2023 Jan-. Available from: https://www.ncbi.nlm.nih.gov/books/NBK493173/
- Liu J, Geng X, Hou J, Wu G. New insights into M1/M2 macrophages: key modulators in cancer progression. Cancer Cell Int. 2021;21(1):1–7.
2) Answer: As described by Wynn (2011), “Fibrosis develops when the wound is severe, the tissue-damaging irritant persists, or when the repair process becomes dysregulated. Thus, many stages in the wound repair process can go awry and contribute to scar formation, likely explaining the complex nature of pulmonary fibrosis”. So, focus on these stages, the first one is the secretion of inflammatory mediators, followed by inflammation, fibroblast migration, and proliferation, and the last phase is the healing out of control, and there is fibrosis formation. These stages were included in the text to improve the reader's understanding. So, we did not expect a linear increase of all parameters analyzed all the time because each one is influenced by one type of cellular event developed in each stage of fibrosis formation. One important result in our model is collagen deposition, the precursor of fibrosis formation. At 60 d.p.i. it is possible to see an intense collagen deposition (Supplementary Figure 2) with increases in collagen type I and III (Supplementary Figure 4). Hence, the collagen deposition is linearly increased until the endpoint of our experiments. To show the differences between 30 and 60 d.p.i, there are images from the whole lung tissue fragment stained for the collagen types.
The frequency of VOIs in µCT analysis reflects how often they appear in the image. In 30 days is possible to notice a diffuse pattern of the image opacification, while in 60 days, the areas of opacification are more concentrated around the airways, as it is possible to observe in supplementary figure 4, leading to a decreased frequency. Compared with the histological findings, the animals had in 30 days a much more intense cellular infiltrate in tissue, which, as argued before, is a characteristic of inflammation.
In Figure 3H, the percentual of VOIs are similar between 30 and 60 days because the opacity depends on diverse events on the tissue. Regarding tissue response, our protocol for inducing IPF initially triggers an acute inflammatory response, which, if left uncontrolled, can progress into a chronic process. This chronic process is characterized by collagen deposition, a typical feature of fibrotic conditions. However, there are limitations in using CT analysis to diagnose and distinguish between these different stages accurately. While CT can detect variations in tissue opacity compared to X-ray, it cannot provide detailed information about ongoing tissue events. Therefore, while the experiment can identify specific structural tissue alterations, confirming fibrosis at the histological level is not sufficient. This limitation was extensively discussed during the COVID-19 pandemic, where CT was used as an imaging diagnostic tool for identifying COVID-19. While medical doctors determined diffuse inflammatory images, these images could also be confused with other lung inflammatory conditions. Consequently, to confirm the histological nature of fibrotic lesions, a microscopic investigation is recommended to identify the specific hallmarks of fibrosis, as presented in our histology results. For this reason, we made histopathological, immunohistochemical, and μCT analyses. All these results showed different lung responses with a decrease in cellular migration but with increased collagen deposition and the same lung opacity. To clarify this point, we included the following sentence in the revised manuscript:
“It is important to note that while µCT analyses can detect differences in structural opacity compared to X-ray, it is unable to identify specific tissue components. Histological investigations are necessary to accurately determine the presence of tissular components. Therefore, regarding the progression of fibrosis, we cannot confirm the typical deposition of collagen. However, both µCT and histopathological analysis reveal a distinct structural difference between the control and peptide-treated mice.”
3) Answer: We changed the methodology to quantify the colors of immunohistochemistry because de ISP was not expressing the positive stain of the tissue (shown before in this file). To better analyze, we changed the methodology and can see an increase in collagen deposition after 60 d.p.i. We chose to use the number of positive pixels normalized per area of the tissue for this new analysis to show the increase of collagen deposition in the tissue. Furthermore, for the groups we called ToAP3+30 days or ToAP4+30 days, we aimed to check if the features observed in the animals of the groups ToAP3 and ToAP4 would be kept the same or if in some way the control of the inflammatory process wouldn’t be enough to stop the fibrogenesis and, therefore, once the treatment stops the fibrotic process would continue. Moreover, in Figures 1, 2, and Supplementary 4, it is possible to observe the progressive inflammation, collagen deposits, and opacification patterns in animals untreated with the peptides but not the same pattern in treated animals. A thorough analysis of the histopathologic pieces from our pathologists described at the 30th-day p.i. the alveolar spaces beginning to be replaced by conjunctive stroma, with fibroblasts, histiocytes, and lymphocytes. It was also possible to verify the existence of airways of different calibers represented by structures with walls constituted by fibromuscular and internally lined by respiratory pattern ciliated epithelium. Using the Masson trichrome stain, detecting the collagen in the connective stroma that was even thicker around airways than in the last time points was possible. At 60 days p.i. we found that the reorganization of the tissue and collagen prevailed; however, the cellular infiltrate was not so robust. The same conditions are not true for our peptide-treated groups. The pulmonary tissue of these animals is more like the animals at 5 days p.i., which we consider a significant effect of the treatment.
About the percentual of VOIs, we compared only the groups that had the same time after the BLM instillation. And we can see that there are differences between them.
4- Answer: As explained, for the groups we called ToAP3+30 days or ToAP4+30 days, we aimed to check if the features observed in the animals of the groups ToAP3 and ToAP4 would be kept the same or if, in some way, the control of the inflammatory process wouldn’t be enough to stop the fibrogenesis and, therefore, once the treatment hampers the fibrotic process would continue. For this reason, the group 60 days is not the most appropriate to compare these results, as we wish to compare them with the treated animals and 30 days, and, therefore, the group 60 days were kept out of the graph of Figure 4 and Figure 6.
5- Answer: The statistical analysis was changed as suggested.
6- Answer: The discussion was modified so it could be more terse.
- Lines 329-330:
Answer: These features were shown in sup 4.
Lines 335-336:
Answer: The histopathological analysis showed an increased immune cell migration until 30 days post-instillation (Supplementary Figure 2). However, if we compare 30- and 60 days post-instillation, there is a difference in tissue structure, with a decrease of cellular infiltration, probably by collagen deposition, as shown in Supplementary Figure 3.
Lines 341-342:
Answer: We carried out experiments in vitro with both peptides, and both decreased TNF production (Veloso Junior et al., 2019). This article’s figure below showed that the peptides decreased TNF and IL-1b cytokines when the macrophages and dendritic cells were stimulated with LPS.
Besides, in our unpublished data on sepsis induced by cecal ligation and puncture, we demonstrated that the levels of serum IL-6 were decreased after intraperitoneal treatment with ToAP3. We also showed that the survival curve is better with the treatment, and the count of cells in BAL is lower than in the non-treated mice. So, based on our previous data, we assumed that in vivo model of PF, the peptides are acting the same way. We explained better the action of peptides based on our published data.
Lines 358-359:
Answered: As described above, the immunomodulatory properties of ToAP3 and ToAP4 are already described in the literature. (Veloso et al., 2019). Also, preventing further tissue damage with peptide treatment, cell infiltration, tissue remodeling, functional impairment, and the switching of cytokine profile from pro-inflammatory to pro-fibrotic are all features that require immunoregulatory properties. The data which refers to morphologic analysis indicating the immunomodulatory action of the peptides is Figure 1, showing a decrease in cellular infiltration, and the treatment with ToAP3 showed a reduction in the gene expression associated with fibrosis formation and inflammation. The analysis of the inflammatory cell phenotype was not the aim of this manuscript.
lines (line 390)
Answered: The sentence is supported by histopathological data that shows leukocyte infiltrate observed in the histopathology and the cytokine profile consistent with the infiltrate.
- The manuscript requires substantial English revision.
Answer: The text was re-edited by a native English speaker
Minor comments
- Pag 2 line 48.
Answer: This information was added to the text
- How did the authors determine the dosage of peptides for the treatment? The authors should report the reasons for the choice in the text.
Answer: This information was added to the text in the Materials and Methods section:
"4.2. Anti-inflammatory peptides treatment
Peptides ToAP3 (FIGMIPGLIGGLISAIK-NH2) and ToAP4 (FFSLIPSLIGGLVSAIK-NH2) [29] were synthesized by FastBio (Ribeirão Preto, SP, Brazil) using solid phase F-MOC strategy (N-9-fluorophenyl methoxy-carbonyl). Peptides analyses were made by Maldi-TOF reflection (MALDI Autoflex Speed TOF/TOF Bruker Daltronics) operated in positive mode with positive reflection; the molecular mass was calculated using Isotope Pattern (version 3.4 Build 76, Bruker Daltonics) as was described in Veloso et al., 2019.
All concentrations chosen for treatment in this work were based previously [28]. After 5 days of BLM instillation, mice were treated intranasally every three days, with 20µL of ToAP3 or ToAP4 in the concentration of 31,25 µM, until the 30th day p.i.. The treatment schedule and evaluations carried out were summarized in Figure 7.”
- In the paragraph “Cytokine quantification” the author should explain what they mean by and how they prepare the macerated lungs (interstitium). Furthermore, they mention the collection of bronchus alveolar lavage fluid (BALF), but no data are reported in this regard.
Answered: The macerated lung information was added to the text in the Materials and Methods section, and the mention of BALF was removed since we did use it to measure cytokines.
4.6. Cytokine quantification
Mice lungs were collected, weighted, and macerated with 1 ml of phosphate-buffered saline (PBS) (interstitium) supernatant were collected after centrifugation at 300g for 5 min and stored at -20ºC. Cytokines TNF-α, IL-1β, IL-10, IL-13, and TGF-β were quantified by ELISA assay Ready-SET-Go!® (eBioscience) following the manufacturer's instructions.
- In Tables 1 and 2 it has been reported “…. fold change >0.5 or >2 and p≤0.05”. It has to be changed in “…. fold change <0.5or >2 and p≤0.05”.
Answer: We corrected the tables' footmarks.
- Authors cite non-English language papers; I recommend replacing them with papers written in English (see ref 23)
Answer: This reference was removed
- The authors should cite more recent references
Answer: References were revised and replaced when it was possible.
- Formatting is uneven (lines 174,175,176; sections 4.4, 4.5, 4.6, 4.7)
Answer: We will check with the journal because we used the template

Round 2
Reviewer 1 Report
The authors thoughtfully replied to all concerns pointed by the Reviewer. Thank you for your diligence.